# Structural polymorphism and diversity of human segmental duplications

Hyeonsoo Jeong[1,2,6], Philip C. Dishuck[1,6], DongAhn Yoo[1,6], William T. Harvey[1], Katherine M. Munson [1], Alexandra P. Lewis[1], Jennifer Kordosky[1], Gage H. Garcia [1], Human Genome Structural Variation Consortium (HGSVC)*, Feyza Yilmaz [3], Pille Hallast [3], Charles Lee [3], Tomi Pastinen [4] & Evan E. Eichler [1,5] ✉

Segmental duplications (SDs) contribute significantly to human disease, evolution and diversity but have been difficult to resolve at the sequence level. We present a population genetics survey of SDs by analyzing 170 human genome assemblies (from 85 samples representing 38 Africans and 47 non-Africans) in which the majority of autosomal SDs are fully resolved using long-read sequence assembly. Excluding the acrocentric short arms and sex chromosomes, we identify 173.2 Mb of duplicated sequence (47.4 Mb not present in the telomere-to-telomere reference) distinguishing fixed from structurally polymorphic events. We find that intrachromosomal SDs are among the most variable, with rare events mapping near their progenitor sequences. African genomes harbor significantly more intrachromosomal SDs and are more likely to have recently duplicated gene families with higher copy numbers than non-African samples. Comparison to a resource of 563 million full-length isoform sequencing reads identifies 201 novel, potentially protein-coding genes corresponding to these copy number polymorphic SDs.

The first draft sequences of the human genome[1] revealed a surprising degree of high-identity duplications dispersed both interchromosomally and intrachromosomally. These SDs have been operationally defined as blocks of homologous DNA greater than 1 kb in length with >90% sequence identity[2]. In humans, ~60% of the pairwise alignments are interspersed, that is, separated by more than 1 Mb within a given chromosome or mapping to non-homologous chromosomes[3,4]. Given their size and high degree of sequence identity, SDs have been some of the last regions of the human genome to be fully resolved[5]. Originally estimated at 5% of the genome, the relative proportion within the telomere-to-telomere (T2T) genome has increased to ~7%, especially as the acrocentric regions of the short arms of human chromosomes have become fully characterized at the sequence level[6,7].

SDs show a wide range of copy number variation in the human species and contribute to structural variation as a result of unequal crossing over (also known as non-allelic homologous recombination (NAHR)). These structural variants contribute to more base-pair differences between humans than those contributed by single-nucleotide variants or indel polymorphisms. Based on sequence read-depth analysis of the short-read sequencing data from the 1000 Genomes Project (1KG)[8], for example, we estimated that 50% of all copy number polymorphisms in the human species that are >1 kb in length map to SDs, which is an approximately tenfold enrichment[9]. Importantly, almost all copy number polymorphic genes in the human species map to these particular regions of the genome[10]. Such copy number polymorphic genes have been strongly implicated in a variety of human

¹Department of Genome Sciences, University of Washington School of Medicine, Seattle, WA, USA. ²Altos Labs, San Diego, CA, USA. ³The Jackson Laboratory for Genomic Medicine, Farmington, CT, USA. ⁴Children's Mercy Hospital and University of Missouri-Kansas City School of Medicine, Kansas City, MO, USA. ⁵Howard Hughes Medical Institute, University of Washington, Seattle, WA, USA. ⁶These authors contributed equally: Hyeonsoo Jeong, Philip C. Dishuck, DongAhn Yoo. *A list of authors and their affiliations appears at the end of the paper. ✉e-mail: ee3@uw.edu

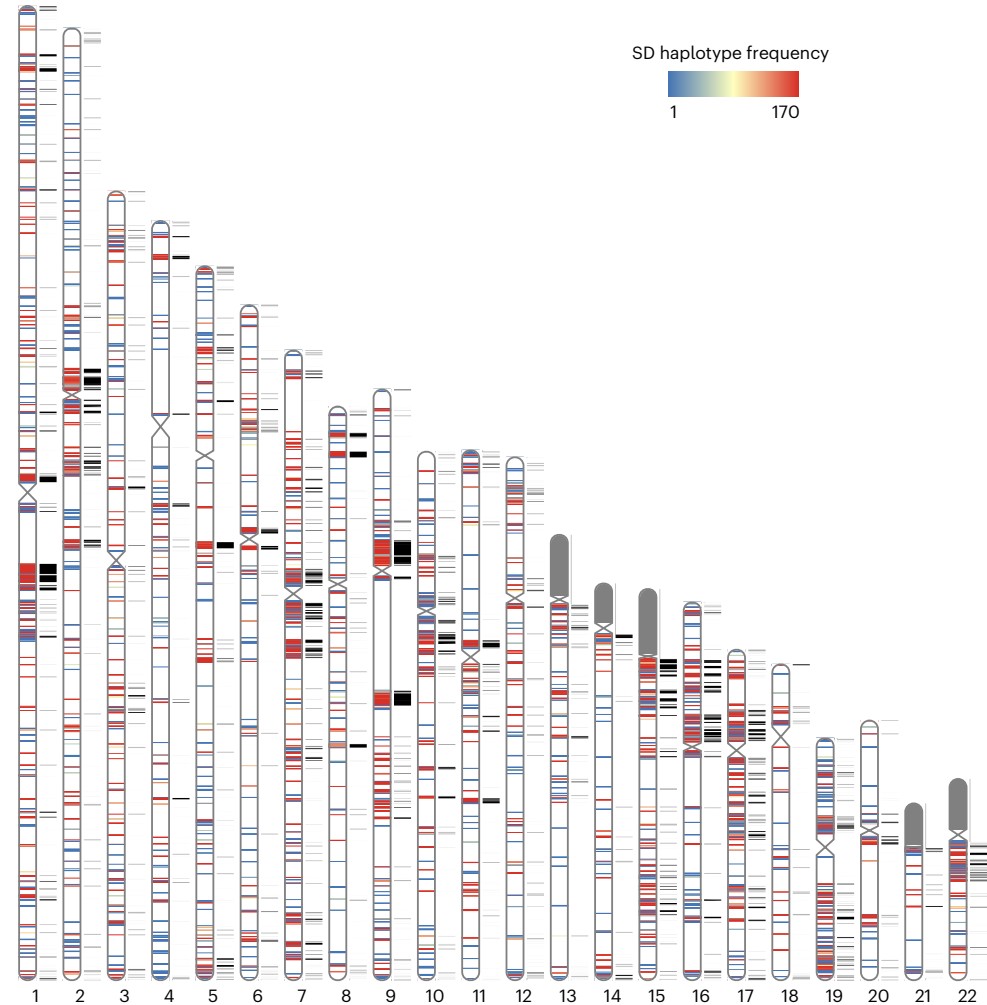

**Fig. 1 | Pangenome representation of human SDs.** Haplotype frequency distribution of intrachromosomal SD content from HPRC and HGSVC haplotype genome assemblies (*n* = 170). SDs are colored by haplotype frequency. SD content on the p-arms of acrocentric chromosomes (chr13, chr14, chr15, chr21 and chr22) was excluded because of assembly errors and potential chromosomal misassignment compared to other autosomal chromosomes. The known SDs of T2T-CHM13 are shown in black next to the ideograms on each chromosome.

diseases, including immune and autoimmune (*FCGR*)[11–13], neurological (*C3/C4*)[14,15] and cardiovascular (*LPA*)[16–18]. More recently, it has become apparent that genes embedded within SDs have had an important role in the evolution of our species, including the expansion of the human frontal cortex (*SRGAP2C*[19], *ARHGAP11B*[20], *TBC1D3* (ref. [21])), adaptation to starch-rich diets (amylase[22,23]) and the development of color vision within the primate lineage (green and red opsins[24]).

Notwithstanding their importance, understanding the genetic diversity of these more complex regions of the genome has been challenging. Most efforts have focused on estimating copy number by mapping short-read data back to a singular reference to discover copy number variant regions[25–27]. Such short-read investigations are useful but incomplete with respect to the genetic characterization of these loci. For example, read-depth analyses can be used to accurately estimate copy number differences in a diploid genome; however, they provide limited information about the location or structure of the duplicated genes or the structure of the associated copy number variants. Similarly, although actual protein-coding differences can be inferred from short-read alignments, these differences are not readily phased, especially in high-identity SDs; thus, genes cannot be fully reconstructed, limiting the potential to distinguish pseudogenes from genes. Finally, mapping short reads to a reference genome introduces reference bias because until recently, the human reference genome was incomplete, with gaps enriched precisely over the most

duplicated regions. Advances in long-read sequencing technology over the past 4 years have addressed these limitations by allowing high-identity regions to be fully phased and assembled, allowing the haplotype, structure and gene annotation to be investigated, in many cases for the first time in the human population[6,28,29]. In particular, the development of PacBio HiFi (high-fidelity) sequencing technology and associated assembly algorithms[30,31] has meant that most SD regions can be fully sequence-resolved at the haplotype level. In this study, we sought to investigate the population genetic diversity of SDs by focusing on 170 human genome assemblies for which HiFi sequence data had been collected as part of the Human Pangenome Reference Consortium (HPRC) and Human Genome Structural Variation Consortium (HGSVC)[10,32].

## Results

### Distribution of shared versus polymorphic SDs

In this study, we analyzed 170 independent genome assemblies and identified SDs (>1 kb and >90%) from 85 human specimens representing 38 African and 47 non-African samples (Supplementary Tables 1 and 2). To investigate how autosomal SD patterns vary among human genomes, we mapped SDs back to the T2T human reference genome (T2T-CHM13), classifying events as either known or new with respect to that reference and then assessed whether they were shared or variable among the 170 human haplotypes. We used these data to estimate the

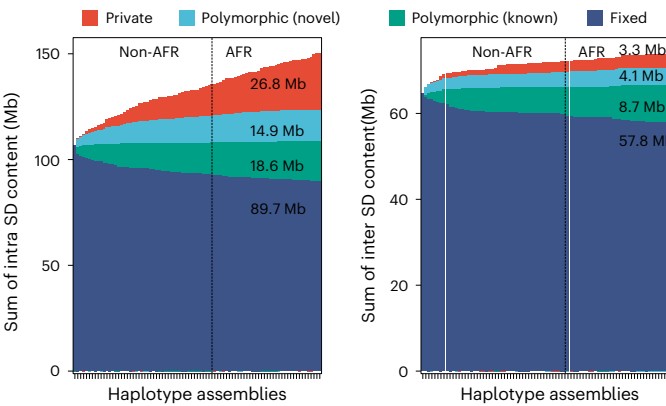

**Fig. 2 | Cumulative sum of SDs by frequency.** Bar plot displaying the cumulative sum of SD content by adding genomes (from left to right) for intrachromosomal and interchromosomal SDs. Four SD frequency categories are considered: 'fixed' are SDs present in all 170 human genome assemblies (that is, conserved in all samples); 'polymorphic (known)' are SDs in the reference genome (T2T-CHM13) that are not fixed; 'polymorphic (novel)' refers to SDs observed in two or more HPRC or HGSVC assemblies yet not present in T2T-CHM13; and 'private' is an SD found in one sample. Samples are grouped by non-African (non-AFR) and then African (AFR) genetic ancestry owing to the expected increased diversity among the latter.

allele frequency of interchromosomal and intrachromosomal duplications, creating a pangenome representation of human SDs (Fig. 1). Given the difficulties in both assembly and mapping of acrocentric SDs, we excluded all short arms or acrocentric chromosomes from this analysis. Acrocentric portions of the human genome are almost entirely composed of repetitive sequences; in fact, the largest and most identical duplications map to this portion of the genome. Moreover, ectopic recombination is rampant among these five chromosomes, making reference mapping almost impossible and delineation of interchromosomal and intrachromosomal SDs extremely challenging. Consequently, these are frequently the last portions of the genome to be accurately assembled and sequenced and require the generation of T2T genomes[6,7]. In total, we identified 2,742 intrachromosomal and 4,772 interchromosomal nonoverlapping SD regions, constituting 6.1% of the genome or 173.21 Mb (150.12 Mb and 73.95 Mb for intrachromosomal and interchromosomal SDs, respectively; 50.86 Mb overlapped between intrachromosomal and interchromosomal SDs) based on the genomic coordinates of the T2T-CHM13 genome.

Compared to the T2T-CHM13 human genome, we classify 47.4 Mb overall as newly discovered, with an estimated rate of accumulation of 408.3 kb SDs being added for each additional sequenced and assembled human genome (Fig. 2). The majority of these novel SDs map intrachromosomally (41.7 Mb), although we classify 7.4 Mb as interchromosomal, and a significant fraction (24.6%) of interchromosomal SDs map to subtelomeric and pericentromeric regions of the human genome (odds ratio, 2.39, $P < 0.01$). Overall, a greater fraction of interchromosomal SDs (78.4%; 16.1 Mb of variable vs 57.8 Mb of invariant SD regions) is fixed compared to intrachromosomal events (59.7%; 60.3 Mb of variable vs 89.7 Mb of invariant SD regions). With respect to intrachromosomal duplications, we find that the majority of novel SDs tend to occur in close proximity to previously known SDs (permutation test, empirical $P < 0.01$), although we do note that certain chromosome arms and regions (for example, p-arms of chromosomes 8, 10, 16, 17 and 19 and q-arms of chromosomes 1, 15 and 22) show an excess of these rarer SDs (Fig. 1 and Supplementary Table 3). As expected from other structural variation studies, the accumulation of novel SDs shows an asymptotic relationship with increasing sample size, and the accumulation is greater for the additional African samples owing to their overall increased genetic diversity (Fig. 2).

## Sequence properties of polymorphic and rare SDs

Among the polymorphic SDs, we further distinguish two groups: rare SDs observed in up to five human genomes (<3% allele frequency) and common SDs observed between 6–20 times (~3–10% allele frequency). We find that rare SDs tend to be longer and have higher sequence identity between SD pairs than common SDs (permutation test, empirical $P < 0.01$) (Fig. 3a and Supplementary Fig. 1). These features are consistent with a more recent origin for the majority of rare SDs; however, there are still SDs that show high sequence divergence that occur at low frequency in the human population, and these may represent ancient SDs that are being lost (Fig. 3a). Notably, we find that low-frequency SDs also tend to be more distant from known SDs than those with a higher allele frequency in the population, suggesting that the interspersion process characteristic of ape genomes is still ongoing in the human population[33,34].

We also considered the orientation and configuration of singleton (single occurrence in a genome validated by read depth) versus polymorphic (not fixed in all human genome assemblies with allele frequency below 90%) SDs. We find that the vast majority (89.1%) of all SD singletons are clustered irrespective of whether they exist in a direct or an inverted orientation (Fig. 3b and Supplementary Table 4). We also find that the proportion of polymorphic SDs classified as interspersed (SD pairs separated by more than 1 Mb) increases in approximately equal proportions between inverted and directly orientated SDs (Fig. 3b). However, interspersed polymorphic SDs favor an inverted orientation ($P = 9.6 \times 10^{-10}$, odds ratio, 1.99, Fisher's exact test). Examples of the structure of rare SDs in an inverted orientation are shown in Fig. 4.

## Gene content and population differences in copy number

Based on the current gene annotation of the T2T-CHM13 genome assembly, we estimate that there are 1,156 duplicated protein-coding genes (s.d. = 49) per diploid genome. Considering all SDs identified in the 170 human genomes, we estimate that 1,340 protein-coding genes are duplicated to copy number four in at least one sample. Of note, 173 of these correspond to single-copy genes in the T2T-CHM13 reference (Methods and Supplementary Table 5). Most low-frequency SDs are incomplete with respect to their ancestral gene model, often involving a subset of the original exons. We caution, however, that incomplete SDs do not guarantee that the duplicates are pseudogenes[19,20,35].

Next, we considered multicopy SDs based on gene content, grouping 1,095 multicopy genes in the T2T-CHM13 reference into 314 gene families. As a control for potential assembly artefacts, we orthogonally evaluated gene copy by correlating ($R^2 = 0.94$) assembly gene copy by predicted copy number from Illumina short-read sequencing read depth (Supplementary Fig. 2). We applied the index of dispersion as a metric of the level of copy number variation for each gene family, which is computed simply as the variance divided by the mean copy number. We identified the 25 most variable gene families in the human genome and contrasted them with the 25 least variable (Fig. 5a,b and Supplementary Fig. 3). As expected, higher copy number gene families (10–50 members) were among the most variable in the human population whereas the most invariant typically were fixed at four or six copies (diploid copy number). The least-variable gene family, *HYDIN*, includes the human-specific duplication *HYDIN2*, which gained neural expression by adopting a new promoter[36]. Similarly, the *RGPD3* family is the eighth least variable based on its copy number and includes two human-specific copies that we find to be under selection[37]. A gene ontology analysis showed that highly variable gene copy associated with female pregnancy (*PSG*), amylase activity (*AMY2*, *AMY1*), immune response (defensin, *KIR2DL*) and unknown biological function whereas fixed copy number SD genes were particularly common among Krüppel-associated box (KRAB) zinc-finger proteins (KRAB-ZFPs) and genes associated with metabolic process (*CYP1*, *CYP2* and *CYP4*) (Supplementary Fig. 4). However, even among high copy number genes, both variable and invariant members are observed.

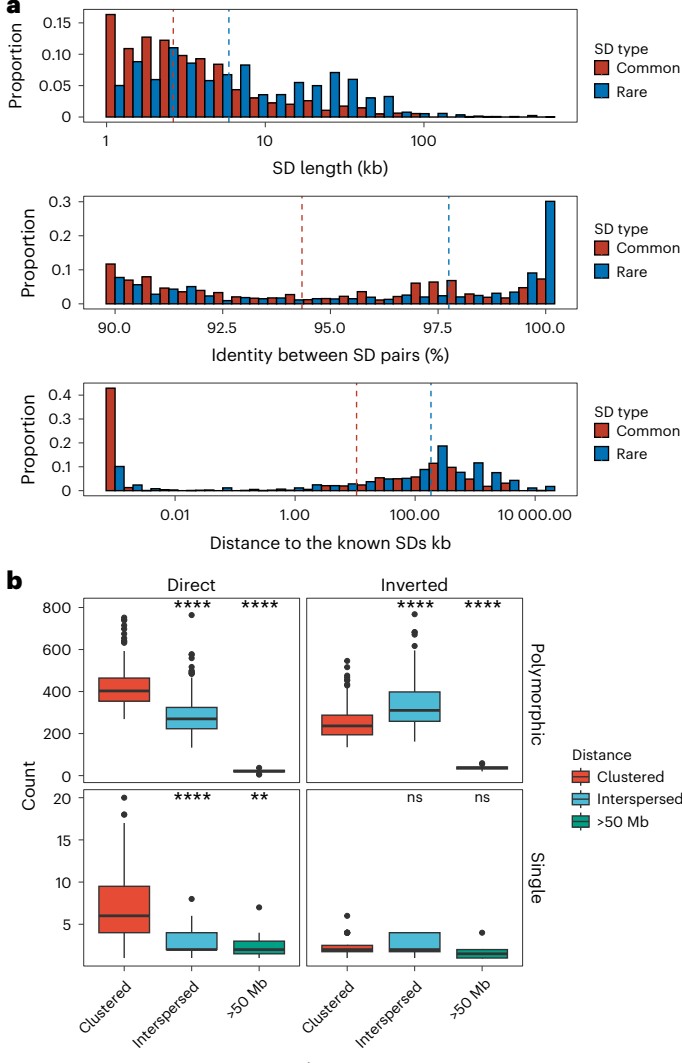

**Fig. 3 | Sequence properties of polymorphic versus rare SDs. a**, Histogram comparing the sequence identity and length of rare and common SDs (see Supplementary Fig. 1 for polymorphic SDs with more subclassified haplotype frequencies). **b**, Orientation and pairwise dispersion of polymorphic and singleton SDs. Each data point represents haplotype assembly (n = 170) and their counts of clustered, interspersed (>1 Mb apart) and distant (>50 Mb apart) SDs. Left and right panels summarize the SDs in direct or inverted orientation, and the top and bottom panels contrast polymorphic versus singleton SDs. The box plot ranges represent the interquartile range (first and third quartile), the horizontal line in each box indicates the median and whiskers indicate data points within 1.5× the interquartile range. In each panel, a two-tailed Wilcox ranked sum test was performed between clustered SDs versus the interspersed or distant SDs; ns, not significant; **P < 0.01; ****P < 0.0001.

Using these highly contiguous assemblies, we are now able to assign copy number polymorphism to specific paralogs in addition to assaying copy number at the level of gene families. For example, for one of the most variable gene families in the human genome (*GOLGA6/8*), we analyzed the copy number variation of each *GOLGA* paralog across our assembled haplotypes (Fig. 5c). To avoid false duplicates, we only included haplotypes with no assembly breaks within 30 kb of the *GOLGA* paralog. Of the named protein-coding paralogs, only *GOLGA6L2*, *GOLGA8M*, *GOLGA8H*, *GOLGA8N* and *GOLGA6B* are fixed at a single copy across all haplotypes, four are variable but never deleted and the remaining 18 are deleted or absent in some haplotypes. This paralog specificity identifies those five single-copy genes as higher-priority candidates for functional analysis, given that they are

fixed in the human population. *GOLGA* paralogs mediate pathogenic microduplications and deletions at 15q11–q13, 15q24 and 15q25, causing forms of intellectual delay including Prader–Willi syndrome[38–42]. Although we observe multi-gene deletions in these regions (Fig. 5c), including genes such as the human-specific fusion gene *CHRFAM7A*, whose deletion has been implicated in Alzheimer's disease pathology[43], none of these deletions extend beyond the SD into the unique critical regions for named syndromes in our samples.

During this gene analysis, we noticed that samples of African ancestry tended to show overall higher copy number for multicopy SDs. We tested this more formally in three ways. First, we compared the intrachromosomal and interchromosomal content irrespective of gene content between genomes of African or non-African origin. Genomes of African origin harbor significantly more intrachromosomal SDs (Mann–Whitney U-test, P = 1.6 × 10⁻⁶) (Fig. 6a and Supplementary Fig. 5). Next, we examined the copy number of gene families by two methods: by counting assembled paralogs in our long-read assemblies and by using read depth to estimate copy number in a larger cohort with short reads (n = 2,196). In the long-read assemblies, we tested the 90 protein-coding gene families with variable copy number (dispersion index, ≥0.1) and mean copy number greater than two for African or non-African samples. In total, 17 gene families showed shifted copy number distribution, and 16 out of 17 showed the same effect by read-depth analysis (Mann–Whitney U-test, Benjamini–Hochberg-corrected P ≤ 0.05). Consistent with the increase in intrachromosomal SDs, for 13 out of 16 gene families (81%), the copy number distribution is higher in African than non-African samples, as shown in Fig. 6b (binomial test, P = 0.01). Finally, with the larger sample of high-coverage Illumina data from unrelated individuals in the 1KG, excluding highly admixed populations (n = 2,196), we considered the 1,171 gene families with a dispersion index of ≥0.1 and mean copy number greater than two in African or non-African samples (Supplementary Fig. 6). Population-differentiated copy numbers are observed in 263 gene families (Mann–Whitney U-test, Benjamini–Hochberg-corrected P < 0.05), with 164 out of 263 (62%) shifted towards higher copy number in African samples (binomial test, P = 0.00004). The gene families with the largest shifts (>15%) are shown in Fig. 6c, with 17 out of 22 (77%) shifted towards higher copy numbers in African samples. All statistically significant gene families are shown in Supplementary Fig. 7. From the assembly test of copy number differentiation, only *GUSBP3* did not replicate in the larger read-depth cohort.

## Genic potential of polymorphic SDs

We sought to assess the transcriptional potential of the structurally polymorphic SDs identified in this study. Given the high degree of sequence identity among the SDs, gene annotation has been difficult with standard RNA sequencing datasets because short reads map equally well to distinct loci. This is especially true for copy number polymorphic genes in which individual copies are >99% identical and can range in copy from 5–40 among different individuals in the population[44]. To address this limitation, we assembled a long-read isoform sequencing (Iso-Seq) resource of 563 million full-length non-chimeric (FLNC) cDNA sequences generated from 241 libraries and 67 distinct tissues (Supplementary Table 6). We mapped each FLNC read both to the T2T-CHM13 genome and a pangenome of 170 human genomes, searching specifically for FLNC reads that mapped better to the pangenome. Specifically, we required at least 99.9% sequence identity to an assembled haplotype and less than 99.7% gap-compressed identity to T2T-CHM13, which is below the expected allelic divergence for most protein-coding regions of the genome. We focused on putative protein-coding genes and constructed 7,081 gene models for cDNA alignments spanning 476 Mb of T2T-CHM13.

We used these reference-divergent cDNA reads that matched better to other assembled haplotypes (n = 1,279,037) to predict protein-coding genes (Fig. 7a). Each additional human haplotype

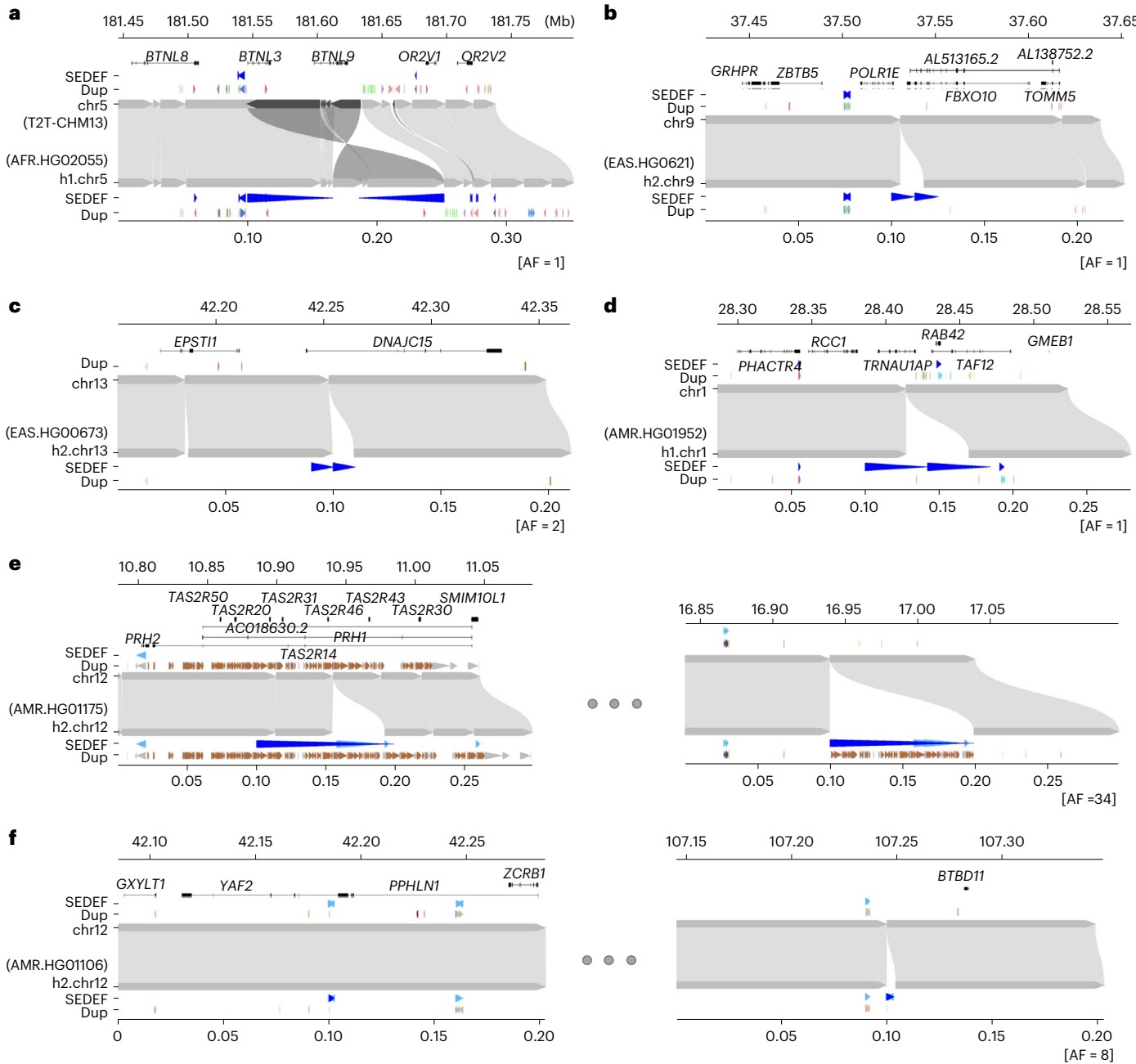

**Fig. 4 | Examples of clustered and interspersed (>1 Mb apart) SDs associated with genes.** In each plot, the top represents the T2T-CHM13 genome aligned to the bottom, new genome assemblies. **a**, Clustered duplication with inverted orientation (65.8 kb; with allele frequency [AF] = 1) found in chr5. **b**–**d**, Clustered and tandem duplications (12.6, 10.3 and 42.3 kb; AFs of 1, 2 and 1, respectively)

in chr9, chr13 and chr1. **e**,**f**, Interspersed duplications of chr 12 (98.9 and 2.5 kb; AFs of 34 and 8) showing duplicated regions in left and right panels. The gene track of the T2T-CHM13 genome assembly is shown at the top, followed by SDs predicted by SEDEF; the respective direction is indicated by blue arrowheads. The DupMasker track shows the duplicon structure.

contained an average of 46 protein-coding gene predictions (range, 13–77) that showed more than 1% divergence from T2T-CHM13 reference annotations, highlighting the importance of additional human genome references to fully assess human genic variation. To count novel gene annotations across haplotypes, we grouped genes and transcripts into gene families by counting only predictions from the haplotype with the greatest number of novel paralogs. This resulted in a total count of 260 putative novel protein-coding genes from 206 gene families. Of these 260 genes, 183 mapped to SD regions, 18 genes mapped to SD regions for at least one sample but not the T2T-CHM13 reference and the remaining 59 genes mapped to unique sequences (not SDs). Gene ontology biological process enrichment analysis of

these genes compared to the background of protein-coding genes within the 476 Mb of sequence examined yielded 13 significantly enriched driver terms, largely related to immunity: positive regulation of leukocyte mediated immunity, antigen processing and presentation of endogenous peptide antigen, rRNA metabolic process, symbiont entry into host, T cell extravasation, regulation of deoxyribonuclease activity, leukocyte cell–cell adhesion, regulation of type II interferon production, regulation of lymphocyte activation, dendritic cell differentiation, detection of bacterium, peptide antigen assembly with MHC class II protein complex and positive regulation of cell–cell adhesion (Benjamini–Hochberg adjusted $P < 0.05$). Twenty of the novel genes belong to the immunoglobulin superfamily[45] and ten are within core

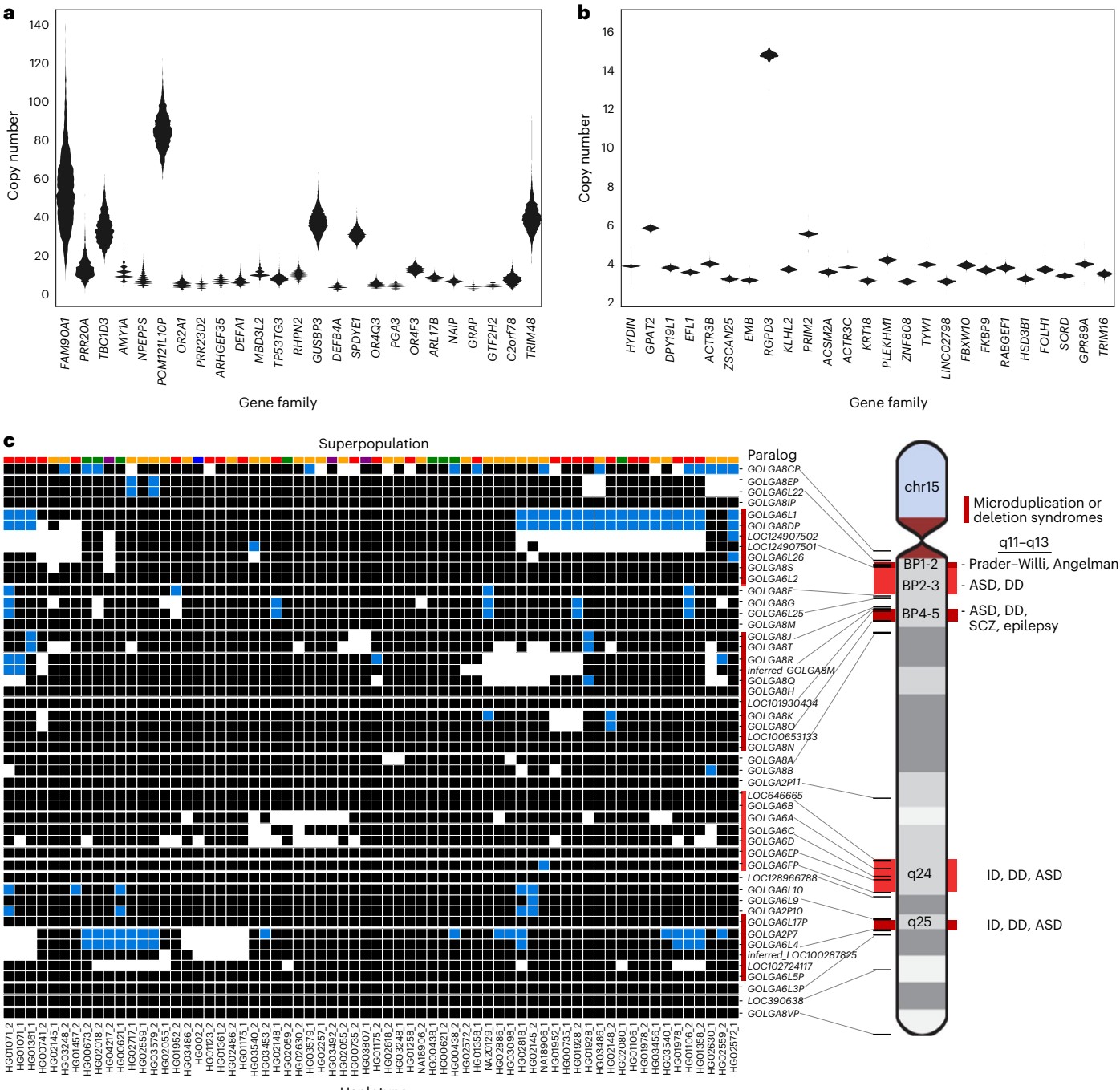

**Fig. 5 | Variable copy number of duplicated genes. a,b,** Gene families with highly variable (**a**) and nearly fixed (**b**) copy numbers are displayed. Gene families are selected and ordered by dispersion index, requiring an average diploid copy number greater than three. Read-depth copy number was estimated with fastCN, using Illumina reads for each sample and comparing it to the T2T-CHM13 genome. **c,** Estimated copy number of *GOLGA6/8* paralogs in each assembled haplotype, based on assembly alignments (white, 0; black, 1; blue, 2). The continental population groups for each haplotype are indicated by color above each column (Africa, gold; East Asia, green; South Asia, purple; Europe, blue; the Americas, red). ASD, autism spectrum disorder; DD, developmental delay; ID, intellectual disability; SCZ, schizophrenia.

duplicons, a group of loci thought to drive the evolution of interspersed larger SD blocks[46] during ape evolution. Only 10.8% (28 out of 260) of these predicted protein sequences had previously been submitted to GenBank.

Notable examples of novel gene annotations include additional copies of *MUC20*, *GSTM*, *TUBB8*, *SIRPB1*, *GOLGA8*, *LRRC37A*, *NBPF1*, *CTAGE* and *UPK3BL1* (Fig. 7b–e)[47–49]. The paralogs with the lowest identity cDNA sequence compared to T2T-CHM13 often have modified

isoform structures, predominately modified amino or carboxy termini owing to the structural rearrangements that led to their formation. For example, the 17q21.31 *KANSL1* inversion haplotype H2 includes a partial *KANSL1* duplication, which acts as an alternate 5′ promoter for *LRRC37A/2*, producing a putatively protein-coding transcript with 39 novel amino acids at its N-terminus followed by sites 870–1,700 of the canonical *LRRC37A/2* isoform (Fig. 7b). The same haplotype also encodes an *NSFP1–LRRC37A2* fusion transcript, which maintains an

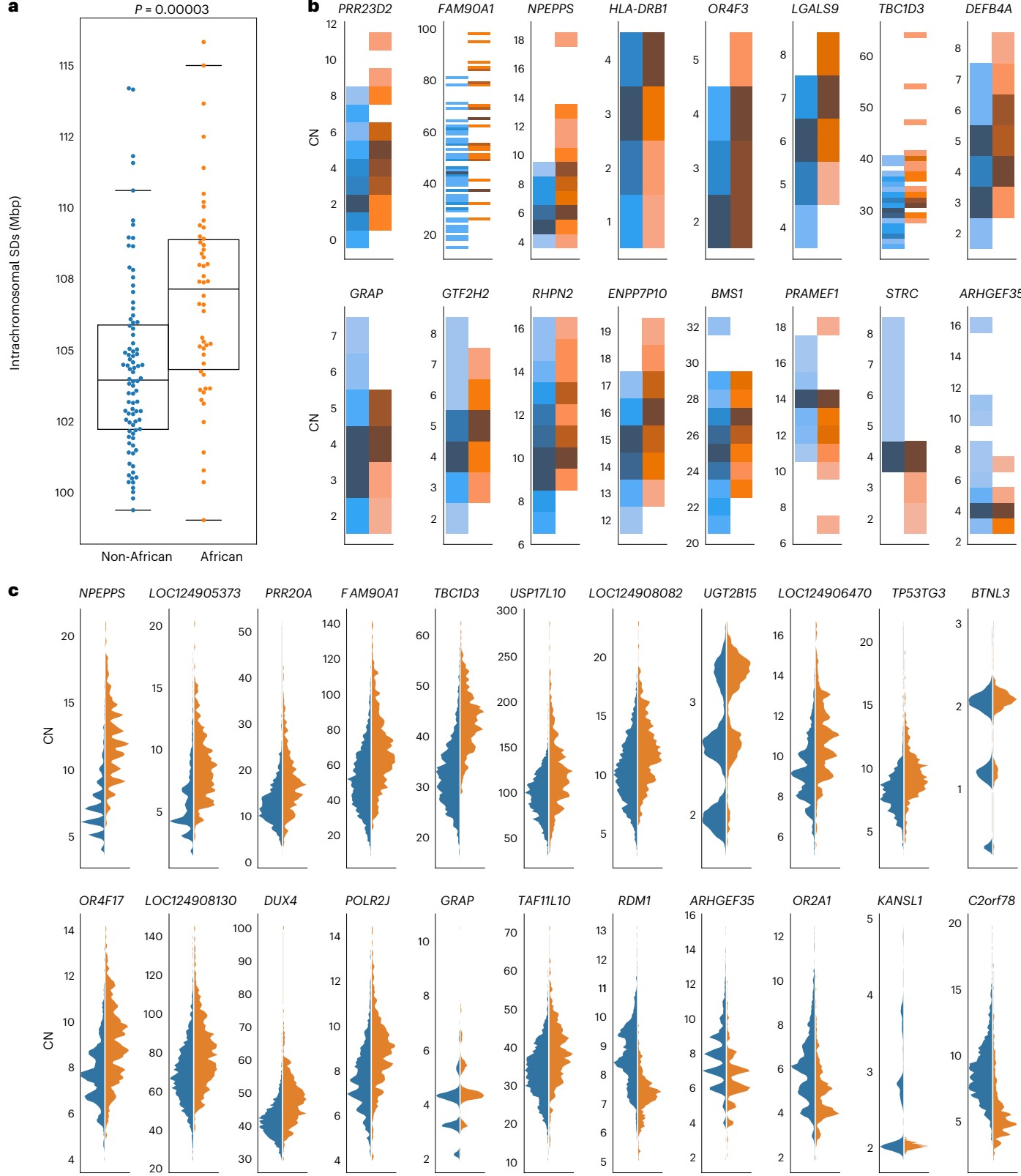

**Fig. 6 | African versus non-African SD copy number variation. a,** Proportion of intrachromosomal SD content between African and non-African populations. African genomes (*n* = 76) have a higher SD content than non-African genomes (*n* = 94), and the difference is significant for intrachromosomal SDs. The box plot ranges represent the interquartile range (first and third quartile), the horizontal line in each box is the median and the whisker indicates the data points within 1.5× the interquartile range. In each panel, a two-tailed Wilcox ranked sum test was performed. **b,** Gene family copy number (CN) variation between populations.

Darker colors represent higher counts of each copy number, normalized per gene. Gene families with significant copy number differences between African and non-African populations are shown (Mann–Whitney *U*-test, Benjamini–Hochberg adjusted *P* < 0.05), excluding *GUSPB3*, which did not replicate in the larger cohort. Gene copy number was estimated from the assemblies by whole-genome alignment; 13 out of 16 gene families average higher copy number in individuals of African ancestry (binomial test, *P* = 0.01). **c,** Gene copy number evaluated by Illumina read depth. The 22 gene families with the largest distribution shift are shown.

open reading frame, predicted to produce a protein with the first 492 amino acids of *NSF* followed by amino acids 41–903 belonging to the core duplicon gene *LRRC37A2* (Supplementary Fig. 8).

We discovered a novel expressed copy of *MUC20* in the paternal haplotype of HG03732 within a 214 kb inversion at chr3q29 that duplicates 73 kb and 37 kb on its edges (Fig. 7c); its expression is supported by full-length cDNA from 14 Iso-Seq libraries from chondrocytes, soft tissue, left colon, induced pluripotent stem cells, human embryonic stem cells and fibroblast cell lines. Structural diversity at 3q29 has been documented in prior work[45], but the expression and gene model of the additional *MUC20* paralog has not yet been reported to our knowledge. Two assembled haplotypes have a complex rearrangement at chr1p36.13 that creates 86 kb of additional sequence compared to T2T-CHM13, including an additional copy of *NBPF1* supported by two Iso-Seq reads from brain tissue and a mammary epithelial cell line (Fig. 7d). In the paternal haplotype of HG01123, an additional copy of *CTAGE* is created by a 16 kb insertion from chr6q23.2 into chr7p35 in the context of a 59 kb duplicated inversion, with expression detected with a single read from each of four Iso-Seq libraries from promyeloblast cells and induced pluripotent stem cells (Fig. 7e). Even among *HLA* genes, whose polymorphisms have been extensively documented owing to their clinical significance, we identified 62 novel alleles across seven distinct genes not currently represented in GenBank or the IPD-IMGT/HLA database (*HLA-A, -B, -C, -G, -DQB1, -DRB1, -DRB5*).

During this analysis, we identified low-identity alignments to *ZNF724* corresponding to a novel KRAB-ZFP absent from the T2T-CHM13 reference. This duplicate gene, provisionally named *ZNF972*, has only 69% identity to its best-matching annotated human gene, *ZNF98* (Fig. 7). *ZNF972* cDNA reads were found in 18 (6%) Iso-Seq libraries and correspond to a 48 kb region within the chr19p12 ZNF cluster, not present in previous human reference genome assemblies (GRCh38 and T2T-CHM13), although it exists in 35.9% of the assembled human haplotypes we analyzed. This region is also present in *Pan*, gorilla and *Pongo* genomes, but an orthologous gene has only been annotated in gorilla as the *ZNF972* coding sequence. Its open reading frame is disrupted in *Pan* and *Pongo* relative to gorilla and human (Fig. 7f,g). Thus, *ZNF972* is an example of an ancestral ape duplicated gene that is still present in gorilla and is present in a subset of humans but probably pseudogenized in other ape lineages.

## Discussion

The last two decades of human genomics research have shown that SDs have an important role in human health and evolution, contributing to genetic diversity, adaptation, genomic instability and susceptibility to disease[19,20,35,50–55]. Despite their importance, understanding how humans vary with respect to this structural feature of our genomes and its potential functional consequence has always been challenging, in large part because the size and high sequence identity of SD repeats have made interrogation of these regions and ~1,000 protein-coding genes mapping within them almost impossible with traditional sequencing and genotyping approaches. As a result, most genome-wide association studies, as well as genome-wide surveys of selection, gene regulation (ENCODE) and transcription (GTEx), have explicitly excluded the most identical SDs from study[56,57]. Even early long-read sequencing-based approaches failed to adequately resolve these particular regions[32,55]. The advent of PacBio HiFi sequencing data[58] along with improved

assembly algorithms has fundamentally changed the calculus[30,31,59]. The sequence accuracy of HiFi data (>99.9%) meant that paralogs and alleles could be fully resolved in a phased genome assembly, making these regions systematically accessible for the first time[10,59,60]. In this study, we took advantage of HiFi data generated as part of the HGSVC and HPRC to analyze SDs in a total of 170 genome assemblies and compared the results to a complete human reference genome (T2T-CHM13). We harmonized the data using the same assembly algorithm and validated copy numbers in individual genomes using Illumina whole-genome sequence data to reveal the location and structure of copy number of SD variation at a population level. Thus, this pangenome representation provides one of the first glimpses of human structural diversity of SDs genome-wide.

Although SDs have been known to be enriched in copy number polymorphisms[26,61,62], the phased genome assemblies allow us to quantify, map and compare this variation and have revealed some unexpected findings. Our analysis of 170 genomes identifies 76.4 Mb of variable (60.3 Mb intrachromosomal and 16.1 Mb interchromosomal) versus 147.5 Mb of invariant (89.7 Mb intrachromosomal and 57.8 Mb interchromosomal) SD DNA; the latter may be more likely to harbor genes that will be functionally constrained[19]. Although fundamentally different in nature, the number of variable nucleotides in this 6% of the genome in these 85 individuals is comparable to the estimated 84.7 million single-nucleotide polymorphisms discovered genome-wide from sequencing the 2,500 individuals from the 1KG[63]. We find that intrachromosomal SDs are twice as likely to be polymorphic compared to interchromosomal SDs, although we should caution that we excluded the acrocentric regions of human short arms from this analysis, where we anticipate rampant ectopic recombination and interchromosomal copy number variation to occur[64]. Although most of the 41.4 Mb of novel SDs (with respect to the finished T2T-CHM13 genome) occur in close proximity to existing regions of SD, we discovered novel sites of interspersed duplications. Such interspersed rare SDs are more likely to be configured in an inverted orientation, minimizing predisposition to large-scale microdeletions but potentially promoting rare inversion polymorphisms in the population[65]. We also note that certain chromosome arms, including chromosomes 1q, 8p, 10p, 15q, 16p, 17p, 19p and 22q, appear enriched for novel SDs, although the basis for this chromosomal bias is unknown.

From a population genetics perspective, it is noteworthy that samples of African ancestry show significantly greater intrachromosomal SD content than 1KG populations belonging to other continental groups. This translates into an overall higher gene copy number for duplicated genes (Fig. 6), an observation we confirmed both by genome assembly and Illumina read-depth analyses (Methods). Although increased variance in copy number would be consistent with the overall 15–20% increase in genetic diversity and greater population substructure that has been reported for populations of African ancestry[66], there are other explanations. Overall higher copy number for duplicated gene families, especially those related to environmental interaction (for example, drug detoxification, immunity), may have provided ancestral human populations with increased genetic diversity in terms of duplicated genes, allowing for selection to operate on different copies to evolve new or modified functions and, therefore, increased fitness. Higher copy number, however, would also lead to greater susceptibility to NAHR-mediated rearrangements with potential negative

---

**Fig. 7 | Discovery of novel genes and transcripts in rare and polymorphic SD regions. a**, 2D histogram display of copy number polymorphic gene families for which FLNC generated from Iso-Seq map better to the pangenome than to the T2T-CHM13 human genome reference. Darker colors represent higher counts of each copy number, normalized per gene. **b**–**e**, Selected haplotypes containing novel gene predictions for *LRRC37A* (**b**), *MUC20* (**c**), *NBPF1* (**d**) and *CTAGE* (**e**) compared to T2T-CHM13 reference where there is FLNC transcript support. Alignment color indicates percent identity. **f**, Comparison of T2T-CHM13 (top)

and HG002 maternal haplotype (bottom) depicts 48 kb polymorphic SD region present in 66 out of 170 haplotypes. Non-human apes all carry a copy of the duplicated sequence. *ZNF* predicted recognition site shown (inset). **g**, Comparison of the novel *ZNF* to its best human match (*ZNF98*, 68% identity) and the most similar existing primate annotation (low-quality protein *ZNF724*-like in gorilla, 95% identity). ProSite-predicted KRAB-ZFP is shown above the sequence.

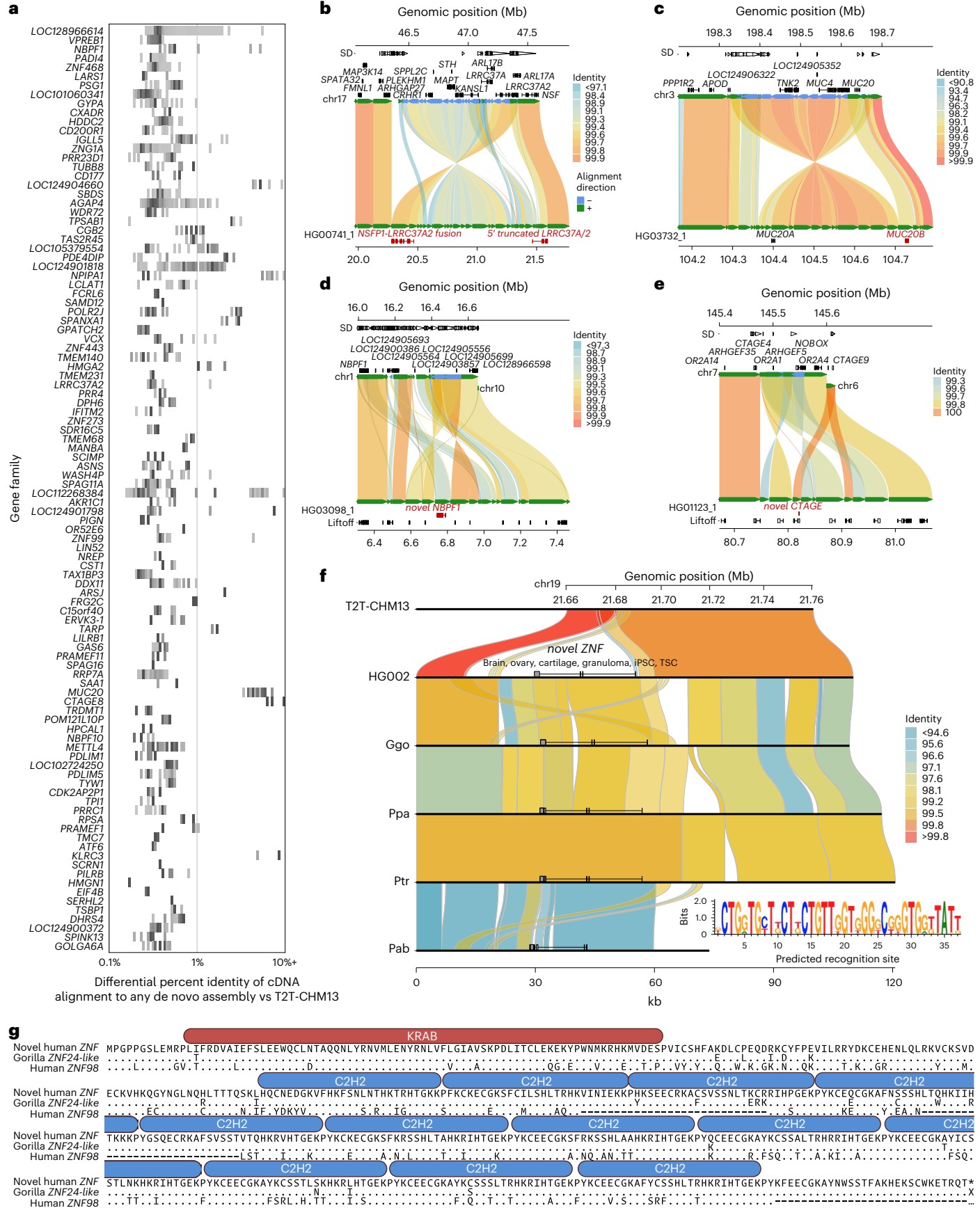

consequences. Alternatively, genetic drift in ancestral populations may have introduced copy number differences, and if the ancestral African populations had higher copy number, mutational biases such as NAHR may have promoted subsequent increases in copy number. It is interesting that read-depth sequencing analysis of some archaic hominins such as Denisova has suggested an overall higher copy number for many gene families compared to modern humans[67].

There are several limitations of the current study. First, we sampled only 85 individuals (170 human genomes), and this represents only a small proportion of potential human genetic diversity. As more human genomes are sequenced and pushed toward T2T status[68], a more complete picture of human genetic diversity will begin to emerge. This will include population-specific paralogs and insights into the mechanisms underlying the formation of interspersed SDs as well as the role of SDs in driving ectopic recombination of acrocentric short arms in the human population[69]. Similarly, our attempt to identify novel genes using a deep resource of human Iso-Seq data should be regarded only as a starting point. The challenge, especially for assessing rarer SDs that harbor duplicated genes, is that full-length cDNA was derived from different individuals than those whose genomes were sequenced and assembled[10,70]. Genomic resources, such as those being generated from the SMaHT (Somatic Mosaicism across Human Tissues) initiative, in which donor-specific assemblies and Iso-Seq data from different human tissues from the same source are gathered, will be required[71,72]. Such matched transcription and assembly data from the same donor will provide a clearer picture of the transcription and tissue specificity of the roughly 1,000 genes mapping to human duplicated sequence. Ultimately, functional characterization will be required to confirm the missing protein-coding copy number polymorphic genes in our genome.

## Online content

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

## Human Genome Structural Variation Consortium (HGSVC)

**Philip C. Dishuck[1,6], DongAhn Yoo[1,6], William T. Harvey[1], Katherine M. Munson[1], Alexandra P. Lewis[1], Jennifer Kordosky[1], Feyza Yilmaz[3], Pille Hallast[3], Charles Lee[3] & Evan E. Eichler[1,5]**

A full list of members and their affiliations appears in the Supplementary Information.

## Methods

### PacBio HiFi sequence production

**University of Washington.** Isolated DNA was sheared using the Megaruptor 3 instrument (Diagenode) twice using settings 31 and 32 to achieve a peak size of ~15–20 kb. The sheared material was processed for SMRTbell library preparation using the Express Template Prep Kit v.2 and SMRTbell Cleanup Kit v.2 (PacBio). After checking for size and quantity, the libraries were size-selected on the Pippin HT instrument (Sage Science) using the protocol '0.75% agarose, 15–20 kb high pass' and a cut-off of 14–15 kb. Size-selected libraries were checked by fluorometric quantitation (Qubit) and pulse-field sizing (FEMTO Pulse). All cells were sequenced on the Sequel II instrument (PacBio) with 30 h video times using version 2.0 sequencing chemistry and 2 h pre-extension. HiFi/CCS analysis was performed using SMRT Link (v.10.1) using an estimated read-quality value of 0.99.

**The Jackson Laboratory.** High-molecular-mass DNA was extracted from 30 million frozen pelleted cells using the Gentra Puregene extraction kit (Qiagen). Purified gDNA was assessed using fluorometric (Qubit, Thermo Fisher Scientific) assays for quantity and FEMTO Pulse (Agilent) for quality. For HiFi sequencing, samples exhibiting a mode size above 50 kb were considered good candidates. Libraries were prepared using the SMRTbell Express Template Prep Kit 2.0 (PacBio). In brief, 12 µl of DNA was first sheared using gTUBEs (Covaris) to target 15–18 kb fragments. Two 5 µg samples of sheared DNA were used for each prep. DNA was treated to remove single-stranded overhangs, followed by DNA damage repair and end repair/A-tailing. The DNA was then ligated with a V3 adaptor and purified using Ampure beads. The adaptor-ligated library was treated with Enzyme mix 2.0 for nuclease treatment to remove damaged or non-intact SMRTbell templates, followed by size selection using Pippin HT (Sage Science), generating a library with a size >10 kb. The size-selected and purified >10 kb fraction of libraries was used for sequencing on the Sequel II (PacBio) system.

We note that ONT data from matched samples generated as part of the HGSVC are available but were generated using ONT R9 flow cells, whereas more recent data from the HPRC and HGSVC are being generated from R10 flow cells. The ONT R9 flow cell generates sequencing reads with an error rate of 2–3% even with the most accurate base-calling model. The high error rate of ONT reads was a major concern for this particular analysis because we wanted to fully characterize highly identical duplicated regions. A hybrid approach using both HiFi and ONT sequencing could increase the continuity of the assembly; however, for the purposes of this study, the HiFi-only-based assembly approach provides sufficient assembly continuity (average contig N50 of 49.59 Mb) and accuracy (quality value > 50), allowing data to be harmonized between HPRC and HGSVC samples.

### Genome assembly and SD annotation

We initially considered a diverse set of 106 human samples (212 haplotype assemblies), all of which originated from the 1KG and for which sufficient HiFi sequence data had been generated as part of previous efforts[10,32]. This included 47 HPRC (all trio binning assemblies using parental short reads) and 53 HGSVC (14 trio and 39 non-trio) samples. We sequenced and assembled all genomes using the same assembly algorithm, hifiasm (v.0.14/v.0.16)[30], which had been shown previously to accurately resolve most (although not all) SD regions[10,65]. We predicted collapses and misassemblies by assessing the read depth of HiFi reads realigned back onto the assemblies using NucFreq implemented in the assembly_eval pipeline (https://github.com/EichlerLab/assembly_eval; Supplementary Table 7). Given the potential for assembly collapse, we further restricted our analysis to 1KG samples for which matched Illumina short-read sequence data were available, and the genomes passed quality control and were all assembled with the same algorithm. SDs are particularly prone to assembly errors or collapses, and this procedure both harmonized the results and allowed

for all duplicated sequences to be validated by Illumina read-depth analysis. We limited SD analyses to the autosomes because of ploidy differences between males and females and the challenges associated with the Y chromosome and pseudoautosomal region in phased assembly. Given the difficulties in mapping acrocentric SDs to specific chromosomes, we also excluded sequence mapping to the short arms of chromosomes 13, 14, 15, 21 and 22. Analysis of these regions will require T2T genome assemblies. The autosomal contigs were scaffolded using RagTag (v.2.1)[73]. We masked repeat content using RepeatMasker (v.4.1) and called SDs using SEDEF[74]. To call SDs, we followed the operational definition of SDs (>90% and >1 kb) from a previous publication[2]. Under neutral evolution, 90% sequence identity allows us to identify SDs that occurred ~35–40 million years ago, and a length threshold >1 kb excludes the effective insertion length of most retrotransposons other than some full-length elements. We matched Illumina short-read sequencing data for all 170 haplotypes, which were used for additional read-depth support of the putative duplicated regions (fastCN)[75].

### Variant calling

We used PAV, an assembly-based phased assembly variant caller, to call variants for 164 genome assemblies, 58 of which were phased into paternal and maternal haplotypes using parental Illumina short-read data (https://github.com/EichlerLab/pav; v.2.1.0). The regions that align 1-to-1 via minimap2 (ref. 76) '-x asm20 –secondary=no -s 25000 -K 8 G', showing no variants, were assigned as the reference, 0|0 genotype, and the regions outside of the alignment blocks were considered missing genotypes when merging the variant calls of individual samples. This was done using BCFtools merge –missing-to-ref followed by BCFtools view with the aligned regions (v.1.9)[77]. Furthermore, to focus on confident variant callset, we additionally defined a 1-to-1 alignment block, which is syntenic with length >1 Mb, in at least 80% of the samples. The population statistics were calculated across this 1-to-1 syntenic, shared alignment blocks of length 2.605 Gb (90%), across the autosomes.

### Iso-Seq and transcript analyses

We used long-read RNA-seq (PacBio Iso-Seq) data to look for evidence of expression of newly discovered low-frequency gene duplications. Examining regions in which ten or fewer haplotypes have a duplication relative to the remainder of the samples, we aligned 563 million FLNC reads from 241 libraries to de novo assemblies and T2T-CHM13 v.2.0 as a reference. Only alignments with >99.9% identity to the novel duplication and <99.7% identity to T2T-CHM13 were considered. To generate gene models for each de novo assembly, we transferred GENCODE v.44 gene models with Liftoff (v.1.6.3)[78], classified Iso-Seq reads compared to the Liftoff gene models with PacBio Pigeon and SQANTI3 (v.5.2)[79] and predicted open reading frame sequences with GeneMark (v.4.3)[80]. Each coding gene prediction was compared to the NCBI non-redundant protein database with BLAST (v.2.15)[81]. We limited our Iso-Seq analysis to transcripts aligning to reference SDs (227.4 Mb), their boundaries (defined as 10% of the SD block size on each edge, 28.1 Mb), SDs seen in at least one assembled haplotype but not the reference (17.9 Mb) and regions corresponding to highly divergent loci in the de novo assemblies (unaligned to T2T-CHM13 with the asm20 preset of minimap2 but forced to align with -r2k,200k -N50 parameters, 202.7 Mb), totaling 476.1 Mb of the T2T-CHM13 genome. Gene ontology was performed with g:Profiler (database e111_eg58_p18_30541362)[82].

### Copy number estimation

**Assembly-based methods.** To estimate copy number, we mapped protein-coding genes (genic sequences from T2T-CHM13) overlapping with SDs to each haplotype assembly using minimap2 (>60% coverage and >90% identity)[76]. Single exons or short genes (coding sequence <200 bp) were excluded. If genes are composed of high repeat content, copy number can be overestimated because of

partial mapping of repeat content. To remove this incorrect mapping, we removed alignments that completely matched the repeat sequence. To increase the contiguity of the alignment and to estimate counts of high copy number genes, we customized the minimap2 options as follows: 'minimap2 -cx map-ont -f 5000 -k15 -w10 -p 0.05 -N 200 -m200 -s200 -z10000 −secondary=yes −eqx'. To avoid any bias caused by switch errors, we estimated gene copy number in diploid genomes. We also excluded duplicate genes found only in one haplotype.

**Illumina fastCN.** Read-depth copy number was estimated with fastCN[75], using Illumina reads for each sample and comparing to the T2T-CHM13 genome. To remove signal from variable number tandem repeats, we excluded fastCN windows that overlapped TRF[83] or WindowMasker[84] calls by more than 10%. To estimate gene copy number, we only considered windows contained within the bounds of each annotated gene, taking the median value. To check for biased copy number estimates, we decomposed the T2T-CHM13 genome into 36-mers and estimated copy number with the fastCN pipeline, simulating fastCN results for a perfectly matched sample and reference. By default, fastCN overcorrected read depth based on GC content for these unbiased artificial reads, owing to over-representation of extreme GC values in the human genome itself. We recalibrated the GC correction, window-by-window, based on these results. Read-depth-based methods also underestimate copy number to a variable extent based on sequence divergence between paralogs, owing to unaligned sequences. To correct for this underestimation, we calculated an adjustment factor for each gene to match fastCN results to alignment-based assembly copy number with T2T-CHM13 as ground truth (Supplementary Fig. 9). Genes that required more than a 50% adjustment to match copy number between methods were excluded from further analysis ($n$ = 82).

Please note that the correlation of determination between Illumina fastCN and assembly copy number differs slightly from that previously reported[85] ($R^2$ = 0.94 vs 0.994). This is because in that previous publication[85], the authors restricted the analysis to 19 large genes and compared Illumina fastCN estimates to $k$-merized assembly fastCN estimates, not quantifying gene copy number in the assemblies directly. In our analysis, we directly quantified the copy number for all SD gene families ($n$ = 314) in the assembled autosomes, excluding p-arms of acrocentric chromosomes and sex chromosomes. Short-read copy number estimates are noisier for the shorter and higher copy number loci, and Illumina fastCN estimates have some residual error from repetitive elements within the genes despite our best efforts to exclude such regions. The authors of the previous publication[85] were able to mitigate this issue because they compared $k$-merized assemblies instead of direct gene copy number estimates.

### Reporting summary
Further information on research design is available in the Nature Portfolio Reporting Summary linked to this article.

### Data availability
The raw sequencing data generated in this study are available under project ID PRJEB58376 (https://www.ebi.ac.uk/ena/browser/view/PRJEB58376) and the HPRC year 1 PacBio HiFi data are available under PRJNA730823 (https://ncbi.nlm.nih.gov/bioproject/PRJNA730823). HPRC genome assemblies are available at https://github.com/human-pangenomics/HPP_Year1_Assemblies. HGSVC genome assemblies used in this study are available at https://eichlerlab.gs.washington.edu/public/HPRC_HGSVC_assemblies. The raw genome sequencing data generated from this study are available at https://ftp.1000genomes.ebi.ac.uk/vol1/ftp/data_collections/HGSVC3. Iso-Seq data generated from this study are available under project ID PRJNA659539 (https://ncbi.nlm.nih.gov/bioproject/ PRJNA659539). T2T-CHM13 (v.2.0) reference genome used in this study is available

under PRJNA559484 (https://www.ncbi.nlm.nih.gov/bioproject/PRJNA559484). All data generated and used in this study were derived from lymphoblastoid cell lines available from the Coriell Institute for Medical Research (https://www.coriell.org) for research purposes. Cell line names are listed in Supplementary Table 1. Source data are provided with this paper.

### Code availability
The custom scripts and pipeline codes in this study are available on GitHub (https://github.com/hrrsjeong/pangenome_SD) and Zenodo (https://doi.org/10.5281/zenodo.11623075)[86].

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

### Acknowledgements
We thank T. Brown for assistance with manuscript editing and preparation. This research was supported, in part, by funding from the National Institutes of Health (NIH) grants R01 HG002385 and R01 HG010169 (to E.E.E.) and U24 HG007497 (to E.E.E. and C.L.). E.E.E. is an investigator of the Howard Hughes Medical Institute (HHMI). This article is subject to HHMI's Open Access to Publications policy. HHMI lab heads have previously granted a non-exclusive CC BY 4.0 license to the public and a sublicensable license to HHMI in their research articles. Pursuant to those licenses, the author-accepted manuscript of this article can be made freely available under a CC BY 4.0 license immediately upon publication.

## Author contributions

H.J., D.Y., P.C.D. and E.E.E. conceived the project. K.M.M., A.P.L., J.K., G.H.G. and T.P. generated sequencing data. F.Y., P.H. and C.L. generated genome assemblies. W.T.H. performed quality-control analyses. H.J. and D.Y. analyzed sequencing data and segmental duplications. P.C.D. analyzed Iso-Seq data. H.J., D.Y., P.C.D. and E.E.E. drafted the manuscript. All of the authors read and approved the manuscript.

## Competing interests

E.E.E. is a scientific advisory board member of Variant Bio. C.L. is a scientific advisory board member of Nabsys and Genome Insight. The other authors declare no competing interests.

## Additional information

**Correspondence and requests for materials** should be addressed to Evan E. Eichler.

# Reporting Summary

## Statistics

For all statistical analyses, confirm that the following items are present in the figure legend, table legend, main text, or Methods section.

| n/a | Confirmed | |
|---|---|---|
| ☐ | ☒ | The exact sample size (*n*) for each experimental group/condition, given as a discrete number and unit of measurement |
| ☐ | ☒ | A statement on whether measurements were taken from distinct samples or whether the same sample was measured repeatedly |
| ☐ | ☒ | The statistical test(s) used AND whether they are one- or two-sided<br>*Only common tests should be described solely by name; describe more complex techniques in the Methods section.* |
| ☒ | ☐ | A description of all covariates tested |
| ☐ | ☒ | A description of any assumptions or corrections, such as tests of normality and adjustment for multiple comparisons |
| ☐ | ☒ | A full description of the statistical parameters including central tendency (e.g. means) or other basic estimates (e.g. regression coefficient) AND variation (e.g. standard deviation) or associated estimates of uncertainty (e.g. confidence intervals) |
| ☐ | ☒ | For null hypothesis testing, the test statistic (e.g. *F*, *t*, *r*) with confidence intervals, effect sizes, degrees of freedom and *P* value noted<br>*Give P values as exact values whenever suitable.* |
| ☒ | ☐ | For Bayesian analysis, information on the choice of priors and Markov chain Monte Carlo settings |
| ☐ | ☒ | For hierarchical and complex designs, identification of the appropriate level for tests and full reporting of outcomes |
| ☒ | ☐ | Estimates of effect sizes (e.g. Cohen's *d*, Pearson's *r*), indicating how they were calculated |

*Our web collection on statistics for biologists contains articles on many of the points above.*

## Software and code

Policy information about availability of computer code

| Data collection | No software was used to collect data. |
|---|---|
| Data analysis | PacBio HiFi data were processed with hifiasm (v0.16), minimap2 (v2.24), RagTag (v2.1), RepeatMasker (v4.1), BCFtools (v1.9). PacBio Iso-Seq data were processed with minimap2 (v2.24), Liftoff (v1.6.3), SQANTI3 (v5.2), GeneMark (v4.3), BLAST (v.2.15). The code for processing PacBio HiFi data is available at Github (https://github.com/hrrsjeong/pangenome_SD) and Zonodo (https://doi.org/10.5281/zenodo.11623075). |

For manuscripts utilizing custom algorithms or software that are central to the research but not yet described in published literature, software must be made available to editors and reviewers. We strongly encourage code deposition in a community repository (e.g. GitHub). See the Nature Portfolio guidelines for submitting code & software for further information.

## Data

Policy information about availability of data

All manuscripts must include a data availability statement. This statement should provide the following information, where applicable:

- Accession codes, unique identifiers, or web links for publicly available datasets
- A description of any restrictions on data availability
- For clinical datasets or third party data, please ensure that the statement adheres to our policy

The raw sequencing data generated in this study are available under project ID PRJEB58376 (https://www.ebi.ac.uk/ena/browser/view/PRJEB58376) and the HPRC

# Research involving human participants, their data, or biological material

Policy information about studies with human participants or human data. See also policy information about sex, gender (identity/presentation), and sexual orientation and race, ethnicity and racism.

| | |
|---|---|
| Reporting on sex and gender | N/A |
| Reporting on race, ethnicity, or other socially relevant groupings | We report analyses of publicly available human genome sequencing data generated by the 1000 Genomes Project (1KG; https:// www.internationalgenome.org/home) and their associated genetic ancestry information, as established and described by the 1000 Genomes Project (https://www.internationalgenome.org/category/population/). |
| Population characteristics | see above |
| Recruitment | see above |
| Ethics oversight | see above |

Note that full information on the approval of the study protocol must also be provided in the manuscript.

# Field-specific reporting

Please select the one below that is the best fit for your research. If you are not sure, read the appropriate sections before making your selection.

☒ Life sciences  ☐ Behavioural & social sciences  ☐ Ecological, evolutionary & environmental sciences

For a reference copy of the document with all sections, see nature.com/documents/nr-reporting-summary-flat.pdf

# Life sciences study design

All studies must disclose on these points even when the disclosure is negative.

| | |
|---|---|
| Sample size | We generated HiFi sequence data from 53 human (14 trio and 39 non-trio) samples. We also analyzed whole-genome assemblies from diverse humans generated by the Human Pangenome Reference Consortium (HPRC). This included 47 trio binning assemblies. In total, we analyzed 170 independent genome assemblies from 85 human specimens representing 38 African and 47 non-African samples. The sample size is sufficient to compare segmental duplication content. |
| Data exclusions | No data were excluded from the analyses. |
| Replication | Independent biological replicates were used for genome assembly as indicated in the manuscript. |
| Randomization | Randomization is not applicable to this study because we did not perform any experiments with treatment or control groups that would necessitate randomization between the subjects. |
| Blinding | Blinding is not applicable to this study because we did not perform any experiments with treatment or control groups that would necessitate blinding. |

# Reporting for specific materials, systems and methods

We require information from authors about some types of materials, experimental systems and methods used in many studies. Here, indicate whether each material, system or method listed is relevant to your study. If you are not sure if a list item applies to your research, read the appropriate section before selecting a response.

## Materials & experimental systems

| n/a | Involved in the study |
|---|---|
| ☒ | ☐ Antibodies |
| ☐ | ☒ Eukaryotic cell lines |
| ☒ | ☐ Palaeontology and archaeology |
| ☒ | ☐ Animals and other organisms |
| ☒ | ☐ Clinical data |
| ☒ | ☐ Dual use research of concern |
| ☒ | ☐ Plants |

## Methods

| n/a | Involved in the study |
|---|---|
| ☒ | ☐ ChIP-seq |
| ☒ | ☐ Flow cytometry |
| ☒ | ☐ MRI-based neuroimaging |

## Eukaryotic cell lines

Policy information about cell lines and Sex and Gender in Research

| | |
|---|---|
| Cell line source(s) | Lymphoblastoid cell lines used for 1KG collection were obtained from the NHGRI Sample Repository at the Coriell Institute for Medical Research. |
| Authentication | Genomic variants in the sequencing data were compared to previously published data. |
| Mycoplasma contamination | All cell lines are negative for mycoplasma contamination. |
| Commonly misidentified lines (See ICLAC register) | No commonly misidentified cell lines were used in this study. |

## Plants

| | |
|---|---|
| Seed stocks | N/A |
| Novel plant genotypes | N/A |
| Authentication | N/A |

