## [Peer Review File · Nature Genetics]

Structural polymorphism and diversity of human segmental duplications

Corresponding Author: Professor Evan Eichler

Version 0:

Decision Letter:

12th August 2024

Dear Evan,

Your Article "Structural polymorphism and diversity of human segmental duplications" has been seen by three referees. You will see from their comments below that, while they find your work of interest, they have raised several relevant points. We are interested in the possibility of publishing your study in Nature Genetics, but we would like to consider your response to these points in the form of a revised manuscript before we make a final decision on publication.

To guide the scope of the revisions, the editors discuss the referee reports in detail within the team, including with the chief editor, with a view to identifying key priorities that should be addressed in revision, and sometimes overruling referee requests that are deemed beyond the scope of the current study. In this case, we ask that you revise the presentation for clarity and accuracy as requested by the reviewers and address the technical points raised by Reviewer #3 related to the quality of the assemblies. We hope that you will find this prioritized set of referee points to be useful when revising your study. Please do not hesitate to get in touch if you would like to discuss these issues further.

We therefore invite you to revise your manuscript taking into account all reviewer and editor comments. Please highlight all changes in the manuscript text file. At this stage, we will need you to upload a copy of the manuscript in MS Word .docx or similar editable format.

*2) If you have not done so already, please begin to revise your manuscript so that it conforms to our Article format instructions, available

[here](http://www.nature.com/ng/authors/article_types/index.html).

*3) Include a revised version of any required Reporting Summary: <https://www.nature.com/documents/hr-reporting-summary.pdf>

Please be aware of our [guidelines](https://www.nature.com/nature-research/editorial-policies/image-integrity) on digital image standards.

Link Redacted

We hope to receive your revised manuscript within 8-12 weeks. If you cannot send it within this time, please let us know.

Nature Genetics is committed to improving transparency in authorship. As part of our efforts in this direction, we are now requesting that all authors identified as 'corresponding author' on published papers create and link their Open Researcher and Contributor Identifier (ORCID) with their account on the Manuscript Tracking System (MTS), prior to acceptance. ORCID helps the scientific community achieve unambiguous attribution of all scholarly contributions. You can create and link your ORCID from the home page of the MTS by clicking on 'Modify my Springer Nature account'. For more information, please visit www.springernature.com/orcid.

Sincerely,
Kyle

Kyle Vogan, PhD
Senior Editor
Nature Genetics
<https://orcid.org/0000-0001-9565-9665>

Referee expertise:

Referee #1: Genetics, structural variation, multi-copy genes

Referee #2: Genetics, structural variation, clinical genetics

Referee #3: Genetics, structural variation, bioinformatics

Reviewers' Comments:

Reviewer #1:
Remarks to the Author:

This paper describes an analysis of human structural variation affecting segmental duplications (SDs) in the human genome, by using a subset of data generated by the Human Pangenome Reference Consortium (HPRC) and Human Genome Structural Variation Consortium (HGSVC), specifically, sequencing reads generated by the PacBio HiFi system.

The authors analyse data from 85 individuals using de novo assembly approaches to identify SDs, and then map sequences from these SDs back to the T2T human reference genome. Orientation, sequence divergence and population frequency are reported and considered. Importantly, gene content and identification of potentially protein-coding transcripts. This latter point is a key novel aspect of the paper, as they use long-read cDNA sequencing to ensure accurate mapping of transcripts to the correct paralogue. Although this is limited to genes that are expressed in lymphoblastoid cell lines, this reveals some new and interesting genes that are similar but distinct to annotated paralogues of disease relevance. For example, the transcripts formed as a result of structural variation at the KANSL1 locus could have particular importance, as SNPs tagging these SVs have been associated with a large number of diseases and traits.

The data analysis seems appropriate and conclusions seem well-supported by the analysis. This will be a very useful paper in the field. I do have some comments that should be addressed:

The introduction could be more focused and up-to-date, there are aspects which are rather vague. The following comments 1-4 are related to this:

1. Line 52. "SDs contribute disproportionately to human genetic diversity". In terms of variable sites, this is incorrect, as there are more SNVs, but correct in terms of bp affected. This needs to be clarified, justified, and references given.
2. Lines 58 to the end of the paragraph. These examples are all valid, but the references are either inappropriate, or not very recent, such that the science has moved on beyond the references supplied. For FCGR, the papers cited are not appropriate, based on small case-control studies using noisy methods, and have not been replicated. The best evidence is (for RA) PMID 27995740, for lung disease/basophil count PMID: 38548989, and a review describing the CNV and disease associations in more detail is PMID 26497510. For LPA, the paper on UK Biobank showing this association should also be

cited PMID: 36779085.

3. Lines 70-74. I sympathise with the authors' overall view here, but the wording is really not appropriate or accurate. Mapping short-reads back to a reference genome has limitations certainly, but the approach is not problematic in itself, as long as we are aware of the limitations – these studies are still useful. In particular, it is simply not the case that they tell you “almost nothing” about the location and structure of the CNV – it shows that at least one copy will be present at the locus on the reference genome, and the overall size of the CNV. They can also tell you something about the protein-coding potential by looking at variable nucleotides within the alignments. This section needs to be rewritten more thoughtfully, emphasising the importance and advantages of long read methods without summarily dismissing other approaches.

4. Lines 76 – 82 Why HiFi data only was included needs to be explicitly explained, in contrast to ONT data which was also generated by the consortium. I think that it is due to the higher quality of the HiFi sequences compared to ONT at that time, and a single assembly approach, but if this is the case this needs to be explained and supported by references.

5. Results line 98. The thresholds used for calling SDs (>1kb, >90% identity) needs to be explained in both technical and biological terms. Why were these chosen, what evolutionary divergence time does this represent, and how does it relate to early SD definitions in papers such as Bailey et al back in 2002?

6. Throughout, kbp and Mbp should be replaced by kb and Mb which is standard style.

7. Lines 252-254. The range figures should come immediately after “average of 46”.

8. Discussion, line 323. SDs don't contribute to selection, but selection is a process that can lead to adaptation, so some SDs are subject to selection. This sentence should be clarified, perhaps by removing “selection”.

9. Line 351. This is related to my comment number 1, but the word “amount” here is doing a lot of heavy lifting and could be misinterpreted. More precision is needed – proportion of variable nucleotides, maybe?

10. Line 360. “Remarkably”. It's not clear to me why this is remarkable, and it is not explained in the paper. Maybe there is negative selection against non-inverted SDs because they can sponsor large chromosomal deletions with deleterious consequences? I'm not sure whether selection would be strong enough, but it is an idea. Is this what the authors are implying here?

11. Line 376-line 377. “greater buffering capacity and adaptive potential”. This concept of evolvability and adaptive potential is controversial – evolution can't look into the future, and more explanation is needed for the concept of “buffering capacity”. The higher copy number might just be drift – Fig6c shows 5/22 show a lower copy number – or it might reflect the action of selection on African populations with larger effective population size. It could also be mutational bias – copy number might be more likely to increase rather than decrease. I would suggest rewriting this section, providing a more balanced view on the explanation between differences in copy number distributions in populations.

12. Line 383 “swath” is not really an appropriate word here, it refers to a strip or belt, originally describing the area of grass cut by a scythe, and retains that rather destructive meaning (in British English at least). “small proportion” is better.

Reviewer #2:

Remarks to the Author:

Jeong, Dishuck, Yoo et al. describe the results of their genome-wide computational analyses of segmental duplications from 170 human haploid genome assemblies originating from 38 African and 47 non-African individuals, excluding short arms of acrocentric chromosomes and sex chromosomes. The long read PacBio HiFi sequences obtained from the Human Pangenome Reference Consortium and Human Genome Structural Variation Consortium were assembled using the hifiasm (v0.14/v0.16) algorithm (PMID: 37165242 and 37164484). To better assess the involvement of SD genes (incl. 201 novel genes) and potential phenotypic consequences, the authors used long-read Iso-Seq resource of full-length non-chimeric cDNA sequences from 241 libraries and 67 distinct tissues.

The manuscript reads well. The rigorously presented data are original and substantially add to the literature. Notwithstanding a number of limitations of their work, discussed in the manuscript, this is a unique and invaluable overview of structural polymorphism and diversity of SDs in humans. However, the manuscript needs some work.

Why were chromosomes X and Y not included?

Please provide more detailed information about the geographic origin of the investigated subjects.

Abstract, line 26

by analyzing 170 human genome assemblies

Please add that they were from 85 individuals, adding the African/non-African breakdown.

Abstract, line 27

where the majority of SDs are fully resolved

What fraction of SDs were not resolved and why?

Abstract, lines 27-28

Excluding the acrocentric short arms, we identify
They should add that sex chromosomes were also not included.

Introduction, lines 42-45

In humans, ~60% of the pairwise alignments are interspersed and separated by more than 1 Mbp within a given chromosome or mapping to the pericentromeric or subtelomeric regions of nonhomologous chromosomes 3,4.
This sentence can be confusing. Why are the interspersed and pericentromeric or subtelomeric mentioned as one category?

Results, lines 86-97

This para belongs better to Methods.

Results, lines 104-105

shared or unique

The term "unique" is unfortunate in this context. It is later (lines 133-134) broken down into rare and common polymorphisms.

Results, lines 107-108

Because of the difficulties in both assembly and mapping of acrocentric SDs,
Please briefly explain these challenges.

Results, lines 107-108

It is unclear what is 24.6% referring to.

Results, lines 138-140

however, we note that highly divergent SDs are still identified that occur at a low frequency in the human population
This statement needs to be rephrased.

Results, lines 142-143

suggesting that the interspersion process characteristic of ape genomes is still ongoing in the human population.
Please provide a reference.

Results, lines 142-143

Using the dispersion index as a metric,
Please define/explain "dispersion index".

Results, lines 199-200

15q11-13, q24, and q25 causing
change to
15q11-q13, 15q24, and 15q25 causing

Results, lines 314-317 and Legend to Fig. 7, line 780

Be consistent with writing Pan, Gorilla, and Pongo – italic or not italic.

Discussion, lines 348-349

identifies 76.4 Mbp of variable versus 147.5 Mbp of invariant SD DNA
These numbers are not mentioned in Results.

Figures 4, 7, and Suppl Fig 5 may be illegible due to a small font.

Reviewer #3:

Remarks to the Author:

The paper reports on population genetic analysis of SDs in 85 human genomes assembled using long-read HiFi data. The haplotype resolved assemblies enable the analysis of inter-chromosomal vs intra-chromosomal SDs and the orientation of rare SD copies. The main findings are greater variation in intra-chromosomal SDs compared to inter-chromosomal SDs, greater intra-chromosomal SDs in African populations and the detection of ~200 novel transcripts with protein-coding potential in SD regions that are not present in current human genome assemblies.

Major comments:

The authors note that SDs are prone to assembly errors and collapses, this is particularly true for long SDs with high sequence identity. It is not clear what is the impact of errors in assembly of particular SDs on the analysis of variation in CN (e.g. Figure 5(A)). In particular, the assemblies were derived from a mix of trio and non-trio samples. If the accuracy of the assemblies in the SD regions is different between the trio and non-trio samples, it could impact the subsequent analysis.

Some statistics on the distribution of African/non-African samples between the trio/non-trio assemblies would be useful.

21 samples were excluded from analysis due to the absence of Illumina short-read data and QC metrics. The correlation between assembly-based gene copy number and short-read read depth-based CN was noted to be 0.94. In a recent paper by the authors (<https://www.nature.com/articles/s41586-023-05895-y>) on a dataset of 102 haplotype assemblies, the correlation was 0.99. It is likely that some SDs have significantly lower correlation/concordance for CN. For these SDs, can the assembly-based analysis of SD copy number variation be considered reliable?

The numbers for the novel protein-coding genes identified seem somewhat discrepant. In the abstract, it is mentioned that 201 novel potentially protein-coding genes were identified. On page 12, this likely corresponds to 183 genes mapped to SD regions and 18 not mapped to CHM13 reference. At the bottom of page 12, a different number (160) is reported. Also, a histogram of the "differential percent identity" for the 201 genes (plotted in Figure 7A) would make it easier to interpret the data.

Line 499: Was the calculation of the adjustment factor for Illumina fastCN previously described or newly developed? If it is new, more details should be provided. I am not sure about the accuracy of a gene-specific adjustment factor calculated using a single assembly.

Version 1:

Decision Letter:

Our ref: NG-A65672R

3rd October 2024

Dear Evan,

Your revised manuscript "Structural polymorphism and diversity of human segmental duplications" (NG-A65672R) has been seen by the original referees. As you will see from their comments below, they are satisfied with the revision, and therefore we will be happy in principle to publish your study in Nature Genetics as an Article pending final revisions to comply with our editorial and formatting guidelines.

We are now performing detailed checks on your paper, and we will send you a checklist detailing our editorial and formatting requirements soon. Please do not upload the final materials or make any revisions until you receive this additional information from us.

Thank you again for your interest in Nature Genetics. Please do not hesitate to contact me if you have any questions.

Sincerely,
Kyle

Kyle Vogan, PhD
Senior Editor
Nature Genetics
<https://orcid.org/0000-0001-9565-9665>

Reviewer #1 (Remarks to the Author):

I thank the authors for responding to my comments clearly, and for pointing me to Supplementary Table 6. They have answered my questions satisfactorily.

Reviewer #2 (Remarks to the Author):

The authors responded satisfactorily.

Reviewer #3 (Remarks to the Author):

The authors' response to the comments from the review is comprehensive and satisfactory. I have no further comments.

NG-A65672 RESPONSE TO REVIEWERS' COMMENTS

Reviewers' Comments:

Reviewer #1:

Remarks to the Author:

This paper describes an analysis of human structural variation affecting segmental duplications (SDs) in the human genome, by using a subset of data generated by the Human Pangenome Reference Consortium (HPRC) and Human Genome Structural Variation Consortium (HGSVC), specifically, sequencing reads generated by the PacBio HiFi system.

The authors analyse data from 85 individuals using de novo assembly approaches to identify SDs, and then map sequences from these SDs back to the T2T human reference genome. Orientation, sequence divergence and population frequency are reported and considered. Importantly, gene content and identification of potentially protein-coding transcripts. This latter point is a key novel aspect of the paper, as they use long-read cDNA sequencing to ensure accurate mapping of transcripts to the correct paralogue. Although this is limited to genes that are expressed in lymphoblastoid cell lines, this reveals some new and interesting genes that are similar but distinct to annotated paralogues of disease relevance. For example, the transcripts formed as a result of structural variation at the KANSL1 locus could have particular importance, as SNPs tagging these SVs have been associated with a large number of diseases and traits.

The data analysis seems appropriate and conclusions seem well-supported by the analysis. This will be a very useful paper in the field. I do have some comments that should be addressed:

We thank the reviewer for acknowledging the usefulness of this paper as well as their time and suggestions. We would like to point the reviewer to Supplementary table 6, which shows the breakdown of transcripts by library source from 67 tissues/cell types, including lymphoblastoid, induced pluripotent stem cells, testis, blood, brain, heart, cartilage, thymus, skin, etc.

The introduction could be more focused and up-to-date, there are aspects which are rather vague. The following comments 1-4 are related to this:

1. Line 52. "SDs contribute disproportionately to human genetic diversity". In terms of

variable sites, this is incorrect, as there are more SNVs, but correct in terms of bp affected. This needs to be clarified, justified, and references given.

Thank you for pointing this out, we revised the sentence as follows:

Previous sentence: “SDs contribute disproportionately to human genetic diversity because of their potential to drive unequal crossing over (aka non-allelic homologous recombination or NAHR).”

Revised sentence: “SDs show a wide range of copy number variation in the human species and contribute to structural variation as a result of unequal crossing over (aka non-allelic homologous recombination or NAHR). These structural variants (SVs) contribute to more base-pair differences between humans than those contributed by single-nucleotide variants (SNVs) or indel polymorphisms.

2. Lines 58 to the end of the paragraph. These examples are all valid, but the references are either inappropriate, or not very recent, such that the science has moved on beyond the references supplied. For FCGR, the papers cited are not appropriate, based on small case-control studies using noisy methods, and have not been replicated. The best evidence is (for RA) PMID 27995740, for lung disease/basophil count PMID: 38548989, and a review describing the CNV and disease associations in more detail is PMID 26497510. For LPA, the paper on UK Biobank showing this association should also be cited PMID: 36779085.

We thank the reviewer for the constructive comment. In the revised manuscript, we updated the references with more recent studies, including those suggested by the reviewers.

3. Lines 70-74. I sympathise with the authors' overall view here, but the wording is really not appropriate or accurate. Mapping short-reads back to a reference genome has limitations certainly, but the approach is not problematic in itself, as long as we are aware of the limitations – these studies are still useful. In particular, it is simply not the case that they tell you “almost nothing” about the location and structure of the CNV – it shows that at least one copy will be present at the locus on the reference genome, and the overall size of the CNV. They can also tell you something about the protein-coding potential by looking at variable nucleotides within the alignments. This section needs to be rewritten more thoughtfully, emphasising the importance and advantages of long read methods without summarily dismissing other approaches.

We revised these sentences to be more specific and also provide a more balanced perspective.

Previous sentences: "Such investigations are problematic for two reasons. First, they tell us almost nothing about the location or the structure of the CNVs or the actual

protein-coding potential of the underlying genes. Second, they introduce a reference bias since, until recently, the human reference genome was incomplete—with gaps enriched precisely over the most CNV regions. Advances in long-read sequencing technology over the last four years, however, have made these regions accessible for the first time"

Revised sentences: "Such short-read investigations are useful but incomplete with respect to genetic characterization of these loci. For example, read-depth analyses can be used to accurately estimate copy number differences in a diploid genome; however, they provide limited information about the location or the structure of the duplicated genes or the structure of the associated CNVs. Similarly, while actual protein-coding differences can be inferred from short-read alignments, these differences are not readily phased, especially in high-identity SDs and, thus, genes cannot be fully reconstructed limiting the potential to distinguish pseudogenes from genes. Finally, mapping short reads to a reference genome introduces reference bias since, until recently, the human reference genome was incomplete—with gaps enriched precisely over the most duplicated regions. Advances in long-read sequencing technology over the past four years have addressed these limitations by allowing high-identity regions to be fully phased and assembled allowing the haplotype, structure, and gene annotation to be investigated in many cases for the first time in the human population."

4. Lines 76 – 82 Why HiFi data only was included needs to be explicitly explained, in contrast to ONT data which was also generated by the consortium. I think that it is due to the higher quality of the HiFi sequences compared to ONT at that time, and a single assembly approach, but if this is the case this needs to be explained and supported by references.

The referee is correct. We focused on using a HiFi-only assembly because of its higher accuracy (sufficient to resolve both haplotypes) when compared to ONT and the fact that the technology has been subject to less turnover making it possible to readily harmonize data from HPRC and HGSVC samples.

We added the following more detailed explanation to the Methods:

"We would like to note that ONT data from matched samples generated as part of the HGSVC are available but were generated using ONT R9 flow cells while more recent data from the HPRC and HGSVC are being generated from R10 flow cells. The ONT R9 flow cell generates sequencing reads with an error rate of 2-3% even with the most accurate base-calling model. The high error rate of ONT reads was a major concern for this particular analysis because we wanted to fully characterize highly identical duplicated regions. A hybrid approach using both HiFi and ONT sequencing could increase the continuity of the assembly; however, for the purposes of this study, the HiFi-only-based assembly approach provides sufficient assembly continuity (average

contig N50 of 49.59 Mb) and accuracy (QV >50) allowing data to be harmonized between HPRC and HGSVC samples.”

5. Results line 98. The thresholds used for calling SDs (>1kb, >90% identity) needs to be explained in both technical and biological terms. Why were these chosen, what evolutionary divergence time does this represent, and how does it relate to early SD definitions in papers such as Bailey et al back in 2002?

We added the information to the Methods based on the reviewer’s suggestion:

“To call SDs, we followed the operational definition of SDs from Bailey et al., 2001. Under neutral evolution, 90% sequence identity allows us to identify SDs that occurred ~35-40 million years ago and a length threshold >1 kb excludes the effective insertion length of most retrotransposons other than some full-length elements.”

6. Throughout, kbp and Mbp should be replaced by kb and Mb which is standard style.

All kbp, Mbp, and Gbp instances have been adjusted to kb, Mb, and Gb throughout to be consistent with *Nature Genetics* style.

7. Lines 252-254. The range figures should come immediately after “average of 46”.

The range figures have been added in the revised manuscript:

“Each additional human haplotype contained an average of 46 protein-coding gene predictions (range 13–77) that showed more than 1% divergence from T2T-CHM13 reference annotations...”

8. Discussion, line 323. SDs don’t contribute to selection, but selection is a process that can lead to adaptation, so some SDs are subject to selection. This sentence should be clarified, perhaps by removing “selection”.

We thank the reviewer for this constructive comment. In the revised manuscript, we removed “selection” in the sentence as the reviewer suggested:

“The last two decades of human genomics research have shown that SDs play an important role in human health and evolution, contributing to genetic diversity, adaptation, genomic instability, and susceptibility to disease....”

9. Line 351. *This is related to my comment number 1, but the word “amount” here is doing a lot of heavy lifting and could be misinterpreted. More precision is needed – proportion of variable nucleotides, maybe?*

As suggested by the reviewer, we revised the sentence by using “the number of variable nucleotides” instead of “amount” to avoid misinterpretation:

“Although fundamentally different in nature, the number of variable nucleotides in this 6% of the genome in these 85 individuals is comparable to the estimated 84.7 million single-nucleotide polymorphisms discovered genome-wide from sequencing the 2,500 individuals from the 1KG.”

10. Line 360. *“Remarkably”. It’s not clear to me why this is remarkable, and it is not explained in the paper. Maybe there is negative selection against non-inverted SDs because they can sponsor large chromosomal deletions with deleterious consequences? I’m not sure whether selection would be strong enough, but it is an idea. Is this what the authors are implying here?*

We agree with the reviewer's point and removed “Remarkably” from the sentence to avoid ambiguity:

“Such interspersed rare SDs are more likely to be configured in an inverted orientation minimizing predisposition to large-scale microdeletions although potentially promoting rare inversion polymorphisms in the population.”

11. Line 376-line 377. *“greater buffering capacity and adaptive potential”. This concept of evolvability and adaptive potential is controversial – evolution can’t look into the future, and more explanation is needed for the concept of “buffering capacity”. The higher copy number might just be drift – Fig6c shows 5/22 show a lower copy number – or it might reflect the action of selection on African populations with larger effective population size. It could also be mutational bias – copy number might be more likely to increase rather than decrease. I would suggest rewriting this section, providing a more balanced few on the explanation between differences in copy number distributions in populations.*

We revised this section as suggested:

“While increased variance in copy number would be consistent with the overall 15-20% increase in genetic diversity and greater population substructure that has been reported for populations of African ancestry, there are other explanations. Overall higher copy number for duplicated gene families, especially those related to environmental interaction (e.g., drug detoxification, immunity) may have provided ancestral human populations with increased genetic diversity in terms of duplicated genes allowing for selection to operate on different copies to evolve new or modified functions and,

therefore, increased fitness. Higher copy number, however, would also lead to greater susceptibility to NAHR-mediated rearrangements with potential negative consequences. Alternatively, genetic drift in ancestral populations may have introduced copy number differences and, if the ancestral African populations had higher copy number, mutational biases such as NAHR may have promoted subsequent increases in copy number.”

12. Line 383 “swath” is not really an appropriate word here, it refers to a strip or belt, originally describing the area of grass cut by a scythe, and retains that rather destructive meaning (in British English at least). “small proportion” is better.

In the revised manuscript, we replaced “swath” with “small proportion” as the reviewer suggested:

“First, we sampled only 85 individuals (170 human genomes) and this represents only a small proportion of potential human genetic diversity.

Reviewer #2:*Remarks to the Author:*

Jeong, Dishuck, Yoo et al. describe the results of their genome-wide computational analyses of segmental duplications from 170 human haploid genome assemblies originating from 38 African and 47 non-African individuals, excluding short arms of acrocentric chromosomes and sex chromosomes. The long read PacBio HiFi sequences obtained from the Human Pangenome Reference Consortium and Human Genome Structural Variation Consortium were assembled using the hifiasm (v0.14/v0.16) algorithm (PMID: 37165242 and 37164484). To better assess the involvement of SD genes (incl. 201 novel genes) and potential phenotypic consequences, the authors used long-read Iso-Seq resource of full-length non-chimeric cDNA sequences from 241 libraries and 67 distinct tissues.

The manuscript reads well. The rigorously presented data are original and substantially add to the literature. Notwithstanding a number of limitations of their work, discussed in the manuscript, this is a unique and invaluable overview of structural polymorphism and diversity of SDs in humans. However, the manuscript needs some work.

We appreciate the constructive comments and careful assessment by the reviewer.

Why were chromosomes X and Y not included?

We chose to focus on autosomes to eliminate potential artifacts in our analysis. First, using HiFi-sequencing data alone, the Y chromosome is extremely difficult to fully assemble, especially within ampliconic regions because there are relatively few unique anchor sequences. An assessment of the Y depends on the construction of T2T genomes, which is currently a major focus of ongoing HPRC efforts. Second, the sex chromosome introduces ploidy biases between males and females making copy number comparison challenging and introducing assembly biases, especially for the X chromosome depending on the location. Even with hybrid-based approaches, the pseudoautosomal regions, for example, remain problematic to fully characterize. Third, all reference mapping was performed originally against a T2T-CHM13 reference lacking the Y chromosome. We have added these points to the Methods to justify our focus on the autosomal portion of these genomes:

“PacBio HiFi sequencing data from 1KG samples were assembled using hifiasm. We limited SD analyses to the autosomes because of ploidy differences between males and females and the challenges associated with Y chromosome and pseudoautosomal region (PAR) in phased assembly. Because of the difficulties in mapping acrocentric SDs to specific chromosomes, we also excluded sequence mapping to the short arms of chromosomes 13, 14, 15, 21 and 22. Analysis of these will require T2T genome assemblies.”

Please provide more detailed information about the geographic origin of the investigated subjects.

We added more detailed information about the geographic origin of the samples used in this study. This information has been updated in Supplementary table 1:

Supplementary table 1. The geographic origin of the samples used in this study.

Source	Sample	Sex	Trio data available	Population	Superpopulation
HPRC	HG01891	Female	1	ACB	AFR
HPRC	HG02257	Female	1	ACB	AFR
HPRC	HG02559	Female	1	ACB	AFR
HPRC	HG02622	Female	1	GWD	AFR
HPRC	HG02630	Female	1	GWD	AFR
HPRC	HG02723	Female	1	GWD	AFR
HPRC	HG02818	Female	1	GWD	AFR
HPRC	HG02886	Female	1	GWD	AFR
HPRC	HG03453	Female	1	MSL	AFR
HPRC	HG03486	Female	1	MSL	AFR
HPRC	HG03516	Female	1	ESN	AFR
HPRC	HG03540	Female	1	GWD	AFR
HPRC	NA18906	Female	1	YRI	AFR
HPRC	NA20129	Female	1	ASW	AFR
HPRC	HG00735	Female	1	PUR	AMR
HPRC	HG00741	Female	1	PUR	AMR
HPRC	HG01071	Female	1	PUR	AMR
HPRC	HG01123	Female	1	CLM	AMR
HPRC	HG01175	Female	1	PUR	AMR
HPRC	HG01361	Female	1	CLM	AMR
HPRC	HG01978	Female	1	PEL	AMR
HPRC	HG02148	Female	1	PEL	AMR
HPRC	HG00438	Female	1	CHS	EAS
HPRC	HG02080	Female	1	KHV	EAS
HPRC	HG02055	Male	1	ACB	AFR
HPRC	HG02145	Male	1	ACB	AFR
HPRC	HG02486	Male	1	ACB	AFR
HPRC	HG02572	Male	1	GWD	AFR
HPRC	HG02717	Male	1	GWD	AFR
HPRC	HG03098	Male	1	MSL	AFR
HPRC	HG03579	Male	1	MSL	AFR
HPRC	HG01106	Male	1	PUR	AMR
HPRC	HG01258	Male	1	CLM	AMR

HPRC	HG01358	Male	1	CLM	AMR
HPRC	HG01928	Male	1	PEL	AMR
HPRC	HG01952	Male	1	PEL	AMR
HPRC	HG00621	Male	1	CHS	EAS
HPRC	HG00673	Male	1	CHS	EAS
HPRC	HG002	Male	1	Ashk	EUR
HPRC	HG03492	Male	1	PJL	SAS
HGSVC	HG02587	Female	0	GWD	AFR
HGSVC	HG03125	Female	0	ESN	AFR
HGSVC	NA19238	Female	0	YRI	AFR
HGSVC	NA19983	Female	0	ASW	AFR
HGSVC	HG00732	Female	0	PUR	AMR
HGSVC	HG01114	Female	0	CLM	AMR
HGSVC	HG01352	Female	1	CLM	AMR
HGSVC	HG02106	Female	1	PEL	AMR
HGSVC	HG00513	Female	0	CHS	EAS
HGSVC	HG00864	Female	0	CDX	EAS
HGSVC	HG02018	Female	1	KHV	EAS
HGSVC	HG02059	Female	1	KHV	EAS
HGSVC	HG00171	Female	0	FIN	EUR
HGSVC	HG00268	Female	0	FIN	EUR
HGSVC	NA12329	Female	0	CEU	EUR
HGSVC	NA12878	Female	0	CEU	EUR
HGSVC	HG03683	Female	0	STU	SAS
HGSVC	NA20847	Female	0	GIH	SAS
HGSVC	HG03807	Female	1	BEB	SAS
HGSVC	HG04036	Female	1	STU	SAS
HGSVC	HG04217	Female	1	ITU	SAS
HGSVC	HG02011	Male	0	ACB	AFR
HGSVC	HG02554	Male	0	ACB	AFR
HGSVC	HG02666	Male	0	GWD	AFR
HGSVC	HG02953	Male	0	ESN	AFR
HGSVC	HG03065	Male	0	MSL	AFR
HGSVC	HG03371	Male	0	ESN	AFR
HGSVC	NA19317	Male	0	LWK	AFR
HGSVC	NA19331	Male	0	LWK	AFR
HGSVC	NA19347	Male	0	LWK	AFR
HGSVC	HG03248	Male	1	GWD	AFR
HGSVC	HG03456	Male	1	MSL	AFR
HGSVC	NA19705	Male	1	ASW	AFR
HGSVC	HG00731	Male	0	PUR	AMR
HGSVC	NA19650	Male	0	MXL	AMR
HGSVC	HG01457	Male	1	CLM	AMR

HGSVC	HG00512	Male	0	CHS	EAS
HGSVC	HG00096	Male	0	GBR	EUR
HGSVC	HG00358	Male	0	FIN	EUR
HGSVC	HG01505	Male	0	IBS	EUR
HGSVC	NA20509	Male	0	TSI	EUR
HGSVC	HG02492	Male	0	PJL	SAS
HGSVC	HG03009	Male	0	BEB	SAS
HGSVC	HG03732	Male	0	ITU	SAS
HGSVC	NA19239	Male	0	YRI	YRI

The population codes for the subpopulation and superpopulation used the information assigned to the samples collected for the 1000 Genomes Project. CHS: Han Chinese South; CDX: Chinese Dai in Xishuangbanna, China; KHV: Kinh in Ho Chi Minh City, Vietnam; CEU: Utah residents (CEPH) with Northern and Western European ancestry; TSI: Toscani in Italia; GBR: British in England and Scotland; FIN: Finnish in Finland; IBS: Iberian populations in Spain; YRI: Yoruba in Ibadan, Nigeria; LWK: Luhya in Webuye, Kenya; GWD: Gambian in Western Division, The Gambia; MSL: Mende in Sierra Leone; ESN: Esan in Nigeria; ASW: African Ancestry in Southwest US; ACB: African Caribbean in Barbados; MXL: Mexican Ancestry in Los Angeles, California; PUR: Puerto Rican in Puerto Rico; CLM: Colombian in Medellin, Colombia; PEL: Peruvian in Lima, Peru; GIH: Gujarati Indian in Houston, TX; PJL: Punjabi in Lahore, Pakistan; BEB: Bengali in Bangladesh; STU: Sri Lankan Tamil in the UK; ITU: Indian Telugu in the UK

Abstract, line 26

by analyzing 170 human genome assemblies

Please add that they were from 85 individuals, adding the African/non-African breakdown.

We added the information to the abstract based on the reviewer's suggestion:

"We present a population genetics survey of SDs by analyzing 170 human genome assemblies (from 85 samples representing 38 Africans and 47 non-Africans) where the majority of **autosomal** SDs are fully resolved using long-read sequence assembly."

Abstract, line 27

where the majority of SDs are fully resolved

What fraction of SDs were not resolved and why?

As mentioned above, this study focused on autosomal SDs excluding the acrocentric short arms and sex chromosomes. Using the T2T as a benchmark of completion, the acrocentric and sex chromosomes would account for 55.3 Mb (27.3 Mb acrocentric and 28.0 Mb of sex chromosome) of SDs leaving 138.4 Mb of SDs as a theoretical

expectation in T2T genomes. In this study, we identified ~121 Mb of SDs per haplotype leaving approximately 17 Mb unaccounted; including 4.5 Mb of putative non-acrocentric collapses based on read depth (shown in table below). As suggested, most of the missing regions correspond to acrocentric regions, the Y chromosome, and other heterochromatic regions that are problematic to assemble. The acrocentric, for example, has the largest and most identical segmental duplications making it difficult for even long-read sequencing to fully resolve.

Supplementary table 7. Summary of misassemblies and collapses within human genomes. The total number of non-acrocentric regions (>1kb in size) and the number of base pairs are summarized.

Sample	Number of Collapse	Collapse bases	Number of Misassembly	Misassembly bases	Source
HG00621.h1	157	5605234	118	1371843	HPRC
HG00621.h2	112	2745195	104	1115476	HPRC
HG00673.h1	247	8797953	115	1245090	HPRC
HG00673.h2	138	3785684	102	1076419	HPRC
HG00735.h1	154	4867526	88	932565	HPRC
HG00735.h2	145	5624387	79	963306	HPRC
HG00741.h1	115	3821209	109	1185968	HPRC
HG00741.h2	129	4385979	110	1253159	HPRC
HG01071.h1	97	3950099	123	2282226	HPRC
HG01071.h2	82	1829159	130	1477330	HPRC
HG01106.h1	182	7349444	82	883891	HPRC
HG01106.h2	119	5135040	62	658943	HPRC
HG01123.h1	145	5212257	130	1520806	HPRC
HG01123.h2	145	4284102	107	1183128	HPRC
HG01175.h1	172	7664884	111	1233676	HPRC
HG01175.h2	107	2967816	107	1097643	HPRC
HG01258.h1	151	4404171	98	1018030	HPRC
HG01258.h2	184	4393731	131	1428052	HPRC
HG01352.h1	151	4752123	109	1141919	HGSVC
HG01352.h2	130	2634847	137	1446214	HGSVC
HG01358.h1	179	5642674	108	1210142	HPRC
HG01358.h2	166	4860239	91	1031615	HPRC
HG01361.h1	98	2479100	88	948745	HPRC
HG01361.h2	200	5838621	104	1087125	HPRC
HG01457.h1	100	1900310	70	716485	HGSVC
HG01457.h2	99	1808642	85	894389	HGSVC
HG01891.h1	124	3799048	65	711279	HPRC
HG01891.h2	143	3879915	94	1007943	HPRC
HG01928.h1	132	5542791	119	1277546	HPRC
HG01928.h2	131	4305551	128	1404510	HPRC
HG01952.h1	143	3911438	92	1170950	HPRC
HG01952.h2	82	1441238	101	1070289	HPRC
HG01978.h1	187	7727963	119	1251765	HPRC
HG01978.h2	117	3884548	113	1326865	HPRC
HG02018.h1	22	376489	176	1864079	HGSVC
HG02018.h2	9	140860	149	1514794	HGSVC

HG02055.h1	191	7046769	62	686916	HPRC
HG02055.h2	162	4995686	56	564707	HPRC
HG02059.h1	100	2301302	124	1321988	HGSVC
HG02059.h2	112	2647514	147	1538074	HGSVC
HG02080.h1	138	4132506	140	1478700	HPRC
HG02080.h2	152	5723599	129	1348465	HPRC
HG02106.h1	132	2917084	153	1567375	HGSVC
HG02106.h2	179	3636185	156	1610777	HGSVC
HG02145.h1	158	3973495	84	854903	HPRC
HG02145.h2	203	5523563	83	858512	HPRC
HG02148.h1	126	3944647	126	1302816	HPRC
HG02148.h2	168	6081572	113	1181408	HPRC
HG02257.h1	180	4781661	79	901278	HPRC
HG02257.h2	157	5643968	63	704532	HPRC
HG02486.h1	167	5234634	70	736841	HPRC
HG02486.h2	120	4058305	63	651177	HPRC
HG02572.h1	235	7178423	162	1805362	HPRC
HG02572.h2	159	4565611	179	1999804	HPRC
HG02622.h1	137	4637847	53	599752	HPRC
HG02622.h2	150	4173323	52	615966	HPRC
HG02630.h1	196	4498053	49	550310	HPRC
HG02630.h2	157	3718532	63	689737	HPRC
HG02717.h1	186	6403195	56	599612	HPRC
HG02717.h2	134	4046748	71	827790	HPRC
HG02723.h1	268	7052963	69	801801	HPRC
HG02723.h2	109	2779830	60	626456	HPRC
HG02818.h1	146	4221203	98	1021792	HPRC
HG02818.h2	153	4540616	103	1080590	HPRC
HG02886.h1	172	6389480	70	787218	HPRC
HG02886.h2	143	4575393	86	957621	HPRC
HG03098.h1	238	7284204	74	858256	HPRC
HG03098.h2	247	7336232	71	787150	HPRC
HG03125.h1	216	5388159	218	2310926	HGSVC
HG03125.h2	187	4717868	203	2146690	HGSVC
HG03248.h1	144	3930157	67	684829	HGSVC
HG03248.h2	101	2534188	54	566957	HGSVC
HG03453.h1	230	5560771	63	757887	HPRC
HG03453.h2	195	5552022	49	514539	HPRC
HG03456.h1	101	2518888	89	963840	HGSVC
HG03456.h2	106	2175516	92	998032	HGSVC
HG03486.h1	215	6218095	78	844184	HPRC
HG03486.h2	193	5019192	64	673117	HPRC
HG03516.h1	127	3333349	87	949821	HPRC
HG03516.h2	150	4190932	90	1002622	HPRC
HG03540.h1	235	5784460	63	777026	HPRC
HG03540.h2	187	6884094	47	496206	HPRC
HG03579.h1	173	5382736	54	591932	HPRC
HG03579.h2	179	4182099	44	478107	HPRC
HG03807.h1	107	1924666	125	1288965	HGSVC
HG03807.h2	104	2171744	138	1455232	HGSVC
HG04036.h1	195	3641160	130	1377967	HGSVC
HG04036.h2	211	4420741	157	1610234	HGSVC

HG04217.h1	124	2818591	121	1239080	HGSVC
HG04217.h2	126	2453813	143	1447432	HGSVC
NA12878.h1	303	7953657	396	4335609	HGSVC
NA12878.h2	313	7938108	440	4696219	HGSVC
NA18906.h1	189	5412124	64	756863	HPRC
NA18906.h2	229	5754690	56	576015	HPRC
NA19705.h1	110	2705689	79	817564	HGSVC
NA19705.h2	72	1700324	61	667387	HGSVC
NA20129.h1	104	2437506	97	1022139	HPRC
NA20129.h2	115	2914825	85	929688	HPRC

Abstract, lines 27-28

Excluding the acrocentric short arms, we identify

They should add that sex chromosomes were also not included.

This information has been added to the revised abstract.

Revised sentence: “Excluding the acrocentric short arms and sex chromosomes, we identify ...”

Introduction, lines 42-45

In humans, ~60% of the pairwise alignments are interspersed and separated by more than 1 Mbp within a given chromosome or mapping to the pericentromeric or subtelomeric regions of nonhomologous chromosomes 3,4.

This sentence can be confusing. Why are the interspersed and pericentromeric or subtelomeric mentioned as one category?

For clarity, we revised the sentence as follows:

“In humans, ~60% of the pairwise alignments are interspersed, i.e., separated by more than 1 Mb within a given chromosome or mapping to nonhomologous chromosomes.”

Results, lines 86-97

This para belongs better to Methods.

Following the reviewer’s suggestion, we revised the paragraph, moving some of it to the Methods section in the revised manuscript.

Previous:

“In this analysis, we initially considered a diverse set of 106 human samples (212 haplotype assemblies), all of which originated from the 1KG and for which sufficient HiFi sequence data had been generated as part of previous efforts. This included 47 HPRC (all trio binning assemblies using parental short reads) and 53 HGSVC (14 trio and 39 non-trio) samples. We sequenced and assembled all genomes using the same

assembly algorithm, hifiasm (v0.14/v0.16), which had been shown previously to accurately resolve most (although not all) SD regions. Because of the potential for assembly collapse, we restricted our analysis to 1KG samples where matched Illumina short-read sequence data were available, and the genomes passed QC and were all assembled with the same algorithm (Methods). SDs are particularly prone to assembly errors or collapses and this procedure both harmonized the results and allowed for all duplicated sequence to be validated by Illumina read-depth analysis. In total, we analyzed 170 independent genome assemblies and identified SDs (>1 kb and >90%) from 85 human specimens representing 38 African and 47 non-African samples (Supplementary table 1).”

Revised:

“In this study, we analyzed 170 independent genome assemblies and identified SDs (>1 kb and >90%) from 85 human specimens representing 38 African and 47 non-African samples (Supplementary tables 1 and 2).”

Results, lines 104-105

shared or unique

The term “unique” is unfortunate in this context. It is later (lines 133-134) broken down into rare and common polymorphisms.

We now use the term “variable” to refer to these SDs that are not fixed in the human population.

Revised sentence: “To investigate how SD patterns vary among human genomes, we mapped SDs back to the T2T human reference genome (T2T-CHM13) classifying events as either known or new with respect to that reference and then assessed whether they were shared or variable among the 170 human haplotypes.”

Results, lines 107-108

Because of the difficulties in both assembly and mapping of acrocentric SDs, Please briefly explain these challenges.

We expanded this paragraph to explain our rationale for excluding acrocentric portions of the human genome:

“Because of the difficulties in both assembly and mapping of acrocentric SDs, we excluded all short arms or acrocentric chromosomes from this analysis. Acrocentric portions of the human genome are almost entirely composed of repetitive sequences—in fact, the largest and most identical duplications map to this portion of the genome. Moreover, ectopic recombination is rampant among these five chromosomes making reference mapping almost impossible and delineation of inter and intrachromosomal SDs extremely challenging. Consequently, these are frequently the last portions of the

genome to be accurately assembled and sequenced and require the generation of T2T genomes (Nurk et al 2022 and Vollger et al. 2022).”

Results, lines 107-108

It is unclear what is 24.6% referring to.

We revised the sentence as follows:

“The majority of these novel SDs map intrachromosomally (41.7 Mb), although we classify 7.4 Mb as interchromosomal, and a significant fraction (24.6%) of interchromosomal SDs map to subtelomeric and pericentromeric regions of the human genome (p -value < 0.01, odds ratio = 2.39).”

Results, lines 138-140

*however, we note that highly divergent SDs are still identified that occur at a low frequency in the human population
This statement needs to be rephrased.*

We revised the sentence as follows:

“however, we note that there are still SDs that show high sequence divergence that occur at low frequency in the human population and these may represent ancient SDs that are being lost.”

Results, lines 142-143

*suggesting that the interspersions process characteristic of ape genomes is still ongoing in the human population.
Please provide a reference.*

In the revised manuscript, we include citations for the relevant papers (Marques-Bonet et al. 2009 and She et al. 2006).

Results, lines 142-143

*Using the dispersion index as a metric,
Please define/explain “dispersion index”.*

The text has been revised to define “dispersion index” as the mean-to-variance ratio. We believe this is superior to relying on variance alone, which overemphasizes the highest copy number gene families.

Revised sentence: “We applied the index of dispersion as a metric of the level of copy number variation for each gene family, which is computed simply as the variance divided by the mean copy number.”

Results, lines 199-200
15q11-13, q24, and q25 causing
change to
15q11-q13, 15q24, and 15q25 causing

As suggested, 15q11-13, q24, and q25 have been adjusted to 15q11-q13, 15q24, and 15q25.

Results, lines 314-317 and Legend to Fig. 7, line 780
Be consistent with writing Pan, Gorilla, and Pongo – italic or not italic.

Non-italicized Pan and Pongo have been adjusted to italicized *Pan* and *Pongo* throughout.

Discussion, lines 348-349
identifies 76.4 Mbp of variable versus 147.5 Mbp of invariant SD DNA
These numbers are not mentioned in Results.

We thank the reviewer for pointing this out. We revised the sentence as follows:

“...identifies 76.4 Mb of variable (60.3 Mb intra and 16.1 Mb interchromosomal) versus 147.5 Mb of invariant (89.7 Mb intra and 57.8 Mb interchromosomal) SD DNA”.

We also updated the sentence in the Results as follows:

Previous sentence: “Overall, a greater fraction of interchromosomal SDs (78.4%) is fixed when compared to intrachromosomal events (59.7%).”

Revised sentence: “Overall, a greater fraction of interchromosomal SDs (78.4%; 16.1 Mb of variable vs. 57.8 Mb of invariant SD regions) is fixed when compared to intrachromosomal events (59.7%; 60.3 Mb of variable vs. 89.7 Mb of invariant SD regions).”

Figures 4, 7, and Suppl Fig 5 may be illegible due to a small font.

We increased the font size of the main and supplementary figures so that all the data figures are now clearly visible in the revised manuscript.

Reviewer #3:

Remarks to the Author:

The paper reports on population genetic analysis of SDs in 85 human genomes assembled using long-read HiFi data. The haplotype resolved assemblies enable the analysis of inter-chromosomal vs intra-chromosomal SDs and the orientation of rare SD copies. The main findings are greater variation in intra-chromosomal SDs compared to inter-chromosomal SDs, greater intra-chromosomal SDs in African populations and the detection of ~200 novel transcripts with protein-coding potential in SD regions that are not present in current human genome assemblies.

We thank the reviewer for taking the time to provide feedback on our paper.

Major comments:

The authors note that SDs are prone to assembly errors and collapses, this is particularly true for long SDs with high sequence identity. It is not clear what is the impact of errors in assembly of particular SDs on the analysis of variation in CN (e.g. Figure 5(A)). In particular, the assemblies were derived from a mix of trio and non-trio samples. If the accuracy of the assemblies in the SD regions is different between the trio and non-trio samples, it could impact the subsequent analysis. Some statistics on the distribution of African/non-African samples between the trio/non-trio assemblies would be useful.

We performed an analysis directly comparing parent–child trios and non-trio samples from African and non-African samples and obtained very similar results. Predictably, data from parent–child trios provided slightly higher statistical significance due to the sample size but both analyses predicted a significant excess of SDs in African versus non-African samples (see figure below).

Supplementary Figure 5. Comparison of SD content in trio vs. non-trio genomes. Both datasets show a significant excess of SD content in African.

As a final check, we also compared our results to recently generated Verkko assemblies from the HGSC (see figure below). These data are unpublished near-T2T assemblies based on hybrid sequencing of HiFi + ONT with pseudo-haplotype phasing via Hi-C and Strand-seq. Analysis of these independent assemblies replicates the excess of the African SDs when compared to non-African SDs (p -value = 0.019, two-tailed Wilcoxon ranked sum test). We also note that Illumina read-depth and gene copy number analyses support this observation. Thus, we believe this observation is very robust.

Response Figure 1. Comparison of SD content in African vs. non-African genomes.

We would like note that to avoid potential assembly bias, Figures 5A and 5B display short-read read-depth copy number estimates from fastCN. We revised the figure legend to clarify the origin of these copy number estimates.

Revised sentence: “**Fig. 5. Variable copy number of duplicated genes.** (A-B) Gene families with highly variable (A) and nearly fixed (B) copy number are displayed. Gene families are selected and ordered by dispersion index, requiring an average diploid copy number greater than three. Read-depth copy number was estimated with fastCN, using Illumina reads for each sample and comparing to the T2T-CHM13 genome.”

21 samples were excluded from analysis due to the absence of Illumina short-read data and QC metrics. The correlation between assembly-based gene copy number and short-read read depth-based CN was noted to be 0.94. In a recent paper by the authors (<https://www.nature.com/articles/s41586-023-05895-y>) on a dataset of 102 haplotype assemblies, the correlation was 0.99. It is likely that some SDs have significantly lower correlation/concordance for CN. For these SDs, can the assembly-based analysis of SD copy number variation be considered reliable?

The referee is correct. In Vollger et al., we reported a higher correlation of determination ($R^2=0.98$ vs. 0.94 in this study) between Illumina read-depth and assembly copy counts. The reason for this is twofold: methodology and sample size. In Vollger et al., we selected 19 loci for analysis and focused only on the largest (and some of the most identical) duplicated loci. In this study, we considered many more genes (1,094 genes grouped into 314 gene families) and, therefore, had much greater variation on the size of loci. Second, Vollger compared k-mer counts from both Illumina WGS reads as well as k-mer-based decomposition of the long-read-based assembled loci. Because we wanted to report copy numbers for individual genes, we compared Illumina WGS k-mers versus the actual counts of our target genes in our study. This leads to reduced accuracy when there are SDs within SDs—such as the NBPF gene family with its DUF1220 tandem repeat domain. Both the gene and this VNTR vary considerably and independently.

We added the following note to the Methods:

“Please note that the correlation of determination between Illumina fastCN and assembly copy number differs slightly from that previously reported (Vollger et al., 2023) ($R^2=0.94$ vs. 0.98). This is because Vollger et al. restricted the analysis to 19 large genes and compared Illumina fastCN estimates to k-merized assembly fastCN estimates, not quantifying gene copy number in the assemblies directly. In this analysis, we directly quantified the copy number for all SD gene families ($n=314$) in the assembled autosomes, excluding p-arms of acrocentric chromosomes and sex chromosomes. Short-read copy number estimates are noisier for the shorter and higher

copy number loci, and Illumina fastCN estimates have some residual error from repetitive elements within the genes despite our best efforts to exclude such regions. Vollger et al. were able to mitigate this issue because they compared k-merized assemblies instead of direct gene copy number estimates.”

The numbers for the novel protein-coding genes identified seem somewhat discrepant. In the abstract, it is mentioned that 201 novel potentially protein-coding genes were identified. On page 12, this likely corresponds to 183 genes mapped to SD regions and 18 not mapped to CHM13 reference. At the bottom of page 12, a different number (160) is reported. Also, a histogram of the "differential percent identity" for the 201 genes (plotted in Figure 7A) would make it easier to interpret the data.

Thank you for catching this. The “201 novel” genes in the abstract is indeed correct and refers to 183 genes in T2T-CHM13 SD regions in addition to 18 genes mapping to SD regions in the assemblies other than T2T-CHM13. The 160 genes on page 12, in contrast, is a typographical error and the correct number should have been $260 = 183 + 18 + 59$ because it includes an additional 59 gene models not mapping to SD regions. Also, as suggested by the reviewer, we have replaced the previous dot plot with a 2D histogram to illustrate the differential % identity (**Fig. 7A**).

Line 499: Was the calculation of the adjustment factor for Illumina fastCN previously described or newly developed? If it is new, more details should be provided. I am not sure about the accuracy of a gene-specific adjustment factor calculated using a single assembly.

This adjustment factor is newly developed as it is only possible to perform with a complete assembly like T2T-CHM13. We agree that as more human genomes become truly complete T2T assemblies, this adjustment factor will become more robust. We updated the Methods to describe this adjustment in more detail.

We added the following text to the Methods:

“This adjustment factor should not be thought of as attempting to correct for any GC biases in Illumina sequencing or library preparation. Instead, the adjustment factor examines the performance of fastCN in a best-case scenario where there is no bias in the Illumina reads used as input for fastCN. To address this, we k-merized the complete T2T-CHM13 human genome to create, in principle, an artificial “perfect” short-read library as input for fastCN. We find that for some regions the inbuilt GC correction for fastCN overcorrects copy number due to the tendency for k-mers at the extremes of GC content to exist at higher frequency in the genome. Our adjustment factor helps to mitigate this overcorrection (**Supplementary Fig. 9**).”

Supplementary Figure 9. Adjustment factor for gene copy number estimation. The read depth of k-mers from decomposed T2T genome assembly is shown as a function of GC composition (blue). Even in a finished genome where there is no experimental or technical error, k-mer read depth is not uniform, since as GC and AT content increases so too does the number of low complexity k-mers mapping elsewhere. Based on this we estimated an adjustment factor required by fastCN (red line) to correct for this bias.